# Recent Review on Biological Barriers and Host–Material Interfaces in Precision Drug Delivery: Advancement in Biomaterial Engineering for Better Treatment Therapies

**DOI:** 10.3390/pharmaceutics16081076

**Published:** 2024-08-16

**Authors:** Rohitas Deshmukh, Pranshul Sethi, Bhupendra Singh, Jailani Shiekmydeen, Sagar Salave, Ravish J. Patel, Nemat Ali, Summya Rashid, Gehan M. Elossaily, Arun Kumar

**Affiliations:** 1Institute of Pharmaceutical Research, GLA University, Mathura 281406, India; rahi18rahi@gmail.com; 2Department of Pharmacology, College of Pharmacy, Shri Venkateshwara University, Gajraula 244236, India; pranshulsethiptl@gmail.com; 3School of Pharmacy, Graphic Era Hill University, Dehradun 248002, India; bhupendra.du@gmail.com; 4Department of Pharmacy, S.N. Medical College, Agra 282002, India; 5Formulation R&D, Alpha Pharma, KAEC, Rabigh 23994, Saudi Arabia; jailanipharmacy@gmail.com; 6National Institute of Pharmaceutical Education and Research (NIPER), Ahmedabad 382355, India; sagarsalave1994@gmail.com; 7Ramanbhai Patel College of Pharmacy, Charotar University of Science and Technology, Changa, Anand 388421, India; ravishpatel.ph@charusat.ac.in; 8Department of Pharmacology and Toxicology, College of Pharmacy, King Saud University, P.O. Box 2457, Riyadh 11451, Saudi Arabia; nali1@ksu.edu.sa; 9Department of Pharmacology & Toxicology, College of Pharmacy, Prince Sattam Bin Abdulaziz University, P.O. Box 173, Al-Kharj 11942, Saudi Arabia; s.abdulrashid@psau.edu.sa; 10Department of Basic Medical Sciences, College of Medicine, AlMaarefa University, P.O. Box 71666, Riyadh 11597, Saudi Arabia; jabdelmenam@mcst.edu.sa; 11School of Pharmacy, Sharda University, Greater Noida 201310, India

**Keywords:** precision therapy, biomaterial, drug delivery, nanotechnology, drug molecule, biological barriers

## Abstract

Preclinical and clinical studies have demonstrated that precision therapy has a broad variety of treatment applications, making it an interesting research topic with exciting potential in numerous sectors. However, major obstacles, such as inefficient and unsafe delivery systems and severe side effects, have impeded the widespread use of precision medicine. The purpose of drug delivery systems (DDSs) is to regulate the time and place of drug release and action. They aid in enhancing the equilibrium between medicinal efficacy on target and hazardous side effects off target. One promising approach is biomaterial-assisted biotherapy, which takes advantage of biomaterials’ special capabilities, such as high biocompatibility and bioactive characteristics. When administered via different routes, drug molecules deal with biological barriers; DDSs help them overcome these hurdles. With their adaptable features and ample packing capacity, biomaterial-based delivery systems allow for the targeted, localised, and prolonged release of medications. Additionally, they are being investigated more and more for the purpose of controlling the interface between the host tissue and implanted biomedical materials. This review discusses innovative nanoparticle designs for precision and non-personalised applications to improve precision therapies. We prioritised nanoparticle design trends that address heterogeneous delivery barriers, because we believe intelligent nanoparticle design can improve patient outcomes by enabling precision designs and improving general delivery efficacy. We additionally reviewed the most recent literature on biomaterials used in biotherapy and vaccine development, covering drug delivery, stem cell therapy, gene therapy, and other similar fields; we have also addressed the difficulties and future potential of biomaterial-assisted biotherapies.

## 1. Introduction

New scientific discoveries about the molecular causes of diseases are leading to the development of more effective treatments, which offers hope for better health outcomes. However, providing affordable and accessible healthcare is becoming increasingly challenging [1]. Precision medicine aims to understand the unique causes of illness in individual patients, allowing for tailored treatments or preventive plans. This approach focuses on creating therapies that benefit specific patient populations [2].

Precision medicine with improving treatment efficacy, while reducing side effects, is the goal of drug delivery systems (DDSs). A DDS has several benefits, including stabilising medications that are prone to instability and helping hydrophobic pharmaceuticals become more soluble in water [3]. Improving the pharmacokinetics and target site accumulation of small compounds, they also aid in controlling medication distribution and activation [4]. DDSs are designed to control drug availability and activity temporally and spatially. They assist in improving the balance between on-target therapeutic efficacy and off-target toxic side effects. Depending on the mode of administration, medicines or drug delivery carriers may encounter various biological obstacles [5].

Instead of barriers, biological barriers should be seen as targetable, interactive, and adaptive interfaces for drug transport. One often-overlooked barrier that must be considered while designing DDSs is patient adherence to medication [6]. Interfacial drug delivery improves drug delivery and activity, depending on the route of administration and implant [7,8]. Innovative interface modification materials and methods have enabled DDS research and performance improvements [9,10,11,12].

Polyethylene glycol (PEG) and polylactic acid (PLGA) are two examples of organic polymers utilised in therapeutics; both have extensive research and application histories in the fields of tissue engineering and delivery systems [13]. The controlled production and inflexible structures of inorganic materials, on the other hand, have attracted a lot of interest. Natural and synthetic biomaterials are further subdivided according to their place of origin. Because of their biodegradability, excellent biocompatibility, low allergenicity, and low toxicity, as well as the fact that they break down into compounds that the host tissues may more readily digest, natural biomaterials have a long history of usage [14]. On the other hand, synthetic materials have more adjustable mechanical qualities and are easy to mass-produce, which makes them more desirable [15]. Because of the promising future of biotherapy in areas including gene therapy, stem cell therapy, tissue engineering, and cancer immunotherapy, it has quickly become a popular area of study [10].

This study reviews recent pre-clinical and clinical studies on biomaterials in biotherapies like stem cell therapy, gene therapy, drug delivery, vaccine delivery, and others, to provide an overview of biomaterial-assisted biotherapies. As a means of medication administration, four distinct systemic drug delivery systems are considered, all of which aim to increase drug accumulation in specific areas, while decreasing the possibility of adverse reactions and off-target effects over long-term use. The in vivo interfaces and biological barriers that a DDS encounters upon oral, IV, and local administration are also addressed in this approach. Additionally, we emphasise how current developments in biomaterial interface manipulation and DDS engineering are significantly enhancing spatially and temporally specific drug treatment.

## 2. Biological Barriers

Nano therapies are limited in their ability to treat inflammatory and cancerous disorders, due to obstacles to medication delivery. Shear force, protein adsorption, and rapid clearance are some of the physiological and biological obstacles that nanoparticles (NPs) must overcome to accomplish biodistribution and medication administration. [16]. Conventional, one-size-fits-all approaches generally fail to overcome these obstacles, because they change in disease states [17,18] (Figure 1). Until most of the biological barriers encountered during administration are overcome through nanocarrier design, site-specific drug delivery will continue to be a challenging goal. While the areas of nanomedicine and nano delivery systems are still in their infancy, a new era in our understanding of NP-based drug delivery will occur when we overcome these barriers and include distinctive design features into the next generation of nanotherapeutics.

## 3. Microenvironmental Barriers

The local microenvironment must be traversed by NPs once they reach the target site. Chemical shifts or physical barriers to entry are two examples of potential obstacles. Therefore, it is essential to have a basic knowledge of the microenvironments that NPs are susceptible to, to engineer them such that they reach the cells or tissues of interest. The physical properties and stability of NPs can be significantly affected by microenvironmental conditions, which differ significantly from circulation conditions. A wide range of acidities and pH values can be found in the gastrointestinal tract [19]. There are enzymes that cause breakdown in the gastrointestinal tract, which makes these conditions more dangerous for many NPs [19,20]. In addition, different diseases might change the gastrointestinal microenvironment in different ways, indicating that people will react differently to biomaterials. Because colon tissue contains varying concentrations of amine surface groups, a study comparing the microenvironments of colon cancer and colitis found that the pathologies led to dendrimer/dextran biomaterial compatibility that depended on the disease [20].

## 4. Barriers to NP Delivery and Biodistribution

The difficulties that NPs face can change with each disease type and its progression, as well as with the method of delivery. Local drug delivery methods help NPs get around some of the problems with systemic medication administration, but they can be more intrusive and require more complicated techniques, which introduces even more limitations. Furthermore, disorders within well-established and easily accessible areas of pathology, including solid tumours or traumatic injuries, may mainly benefit from local delivery. Therefore, NP applications typically involve systemic administration.

The initial stage for NPs in circulation to reach their target tissues is extravasation [21,22]. Size is one of the NP features that can affect extravasation. Compared to big NPs, small NPs typically have an easier time passing through capillary walls [23,24,25]. 

According to Figure 2, the distribution of NPs throughout the organs is size dependant, with the liver and spleen often accumulating the most, from two separate sources [23,26]. Tumour vasculature is one pathological environment that can change size-dependent distribution, because it has bigger-than-normal intercellular spaces, which allow larger NPs to exit the vessels [24]. A translational problem for applications requiring localisation is the non-specific dispersion that results from extravasation [23].

Enhancing biodistribution is possible by the optimisation of the administration route. A lot of research has looked at how different routes of administration affect the fate of NPs, but this is applicable to all drugs, because their fate and effectiveness in vivo can be affected by various factors [27,28]. When administered intravenously, polymeric (poly(lactic-co-glycolic) acid) NPs tend to collect in the liver and spleen. However, when injected subcutaneously or intranodally, the NPs are more likely to gather in local lymph nodes. Some immunotherapeutic uses may benefit from these alternative administration methods, since they allow NPs to reach the lymphatic system before systemic circulation [29,30]. Pulmonary administration, more especially NP inhalation, is another strategy that has been investigated more and more for NP delivery to avoid extravasation. It also avoids hepatic first-pass metabolism. Inhaled NPs have an advantage when it comes to reaching lung tissue, but they still encounter the distinct challenges posed by mucus and pulmonary surfactant. These substances serve as physical barriers to lung delivery, and their specificities differ among patients and diseases [31,32]. With a limited adjustment of the administration route, approved NP formulations are typically given intravenously or intratumourally in clinical settings. These investigations are continuing, even though pre-clinical work is being conducted to investigate potential alternatives. Although there may be ways to improve distribution by choosing the best route of administration for NPs, many of the existing approaches still lead to NPs being dispersed too widely, without the appropriate amount of targeting or specificity.

Tight junctions among the endothelial and epithelial cells of the blood–brain barrier (when administered intravenously) and the gastrointestinal tract (when administered orally) are physical barriers to the distribution of NPs. Endothelial cells lining the blood–brain barrier (BBB) is responsible for NP uptake and exocytosis, the processes by which NPs enter the central nervous system (CNS) [33]. One good method to infiltrate tumour tissue or transport medications to the brain is receptor-mediated transcytosis [34,35]. Endothelial cell plasma membrane transporter variability complicates this BBB-bridging approach. But some transporters, like glucose transporters, are always overexpressed on the BBB, and increasing NP transit across the BBB is possible by targeting common targets, like vascular cell adhesion molecule 1 [36]. It is possible to use these two molecules to transport NPs. The transferrin receptor is one of several potential targeting sites; however, despite its theoretical advantages to other transporter types, it has not yet achieved clinical success [37]. Very little of the NP dose given systemically reaches target cells, and even less enters the central nervous system, when using transferrin receptor systems [33]. In general, the BBB is still a big obstacle for NPs that are given systemically and try to enter CNS regions. Because it gets around the BBB and all the problems with systemic distribution, intranasal administration is being considered more and more to deliver NPs to the brain [38]. However, the intranasal route has encountered significant challenges due to factors such a restricted dosage amount and variables associated with patient congestion and mucus [39,40].

## 5. Nanoparticles (NPs) in Precision Medicine

The development of highly modular and adaptable techniques for therapeutic delivery is necessary to adjust for the great variability in biological barriers and disease states within and across patient groups. In this section, we examine the ways in which different NP qualities affect delivery. We will specifically look at how factors like design, targeting, material properties, and responsiveness can help NPs overcome barriers that are unique to each patient’s disease (Table 1).

### Local NP Distribution

Since solid tumours make NP penetration and stability difficult, the cancer microenvironment has been thoroughly investigated for potential barriers to local dispersion [41]. Interstitial fluid pressure, vasculature, and extracellular matrix (ECM) density are only a few of the tumour microenvironment features that limit the permeation and penetration of NPs [42,43,44,45]. Therefore, there has been a lot of discussion over what exactly causes NPs to accumulate in tumours, and here are just a few of the proven trends that link NP design to tumour delivery. Hydrodynamic diameters greater than 100 nm, rod-shaped topologies, near-neutral charges, and inorganic material compositions are a few of the NP features that can encourage tumour accumulation [24].

An additional important factor in NP destiny determination is the tumour microenvironment. The enhanced permeation and retention (EPR) effect describes how NPs might concentrate in tumours due to the defective and heterogeneous vasculature inside tumours. This is because the tumours’ leaky arteries allow NP extravasation. The part played by the EPR effect in NP buildup in tumours is a matter of controversy. The EPR effect is responsible, according to some research, for the accumulation of up to 10–15% of injected NPs at the tumour site, as opposed to 0.1% of free medication. On the other hand, new research in a mouse cancer model using imaging and computational methods has shown that passive transport, including the EPR effect, can only explain a small portion of the NP buildup in tumours. Rather, the results point to the possibility that molecular processes, immune cell contacts, and protein coronas play a significant role in the accelerated accumulation of NPs in tumours [35].There is evidence from meta-analyses to back up these claims; for example, one study looked at 232 datasets and found that, on average, just 0.7% of injected NP dosages reach tumours, which significantly reduces the significance of the EPR effect. A recent study has shown that quantifying NP distribution using non-standard techniques might lead to deceptive results, which could have caused bias [46]; therefore, it is vital to be aware of the limitations of these generalised findings. Research into the long-term effects of EPR on NP buildup should therefore focus on assessing the validity of the current metrics for measuring distribution and delivery.

Biofilms and mucus are two other obstacles that NPs encounter on their way to local dissemination. The mesh pore size, which can range from 10 to 1000 nm, is determined by the lengths between adjacent polymer linkages within mucus layers; as a result, smaller items are able to permeate through, while bigger ones are retained [44]. Mucus may sort things by size and also capture them by non-specific interactions, which cause them to be quickly washed away from epithelial surfaces. The composition, hydration, and viscoelasticity of mucus vary depending on its physiological location, causing it to function differently even if it serves a comparable purpose across the body [47]. Mucus in the lungs is typically thinner and more adaptable, forming a heterogeneous barrier, in contrast to the adherent, thick coating of mucus in the gastrointestinal system [47].

While mucus behaviour is consistent throughout all of these physiological settings, there are, however, differences and ever-changing barriers within the mucus of certain organ systems. Factors like fibre consumption impact both the thickness of mucus and the turnover rate in the gastrointestinal tract, where it can vary between 40 and 450 μm in the stomach and 110 to 160 μm in the colon [32]. A micrometre-scale pH differential exists across the mucosal barrier’s transition from the almost-neutral endothelial cell surfaces to the acidic intestinal lumen, making it an extremely unstable environment for NP platforms. In gastrointestinal tract illnesses, changes in glycosylation patterns, pH, and mucus layer thickness can alter these mucus properties as well [48].

**Table 1 pharmaceutics-16-01076-t001:** NPs studied for precision medicine therapy or diagnosis.

Formulation	Targeted Molecule	Advancement	Uses	Refs.
LNPs	mRNA	Charge	Cancer and autoimmune immunotherapies	[49]
	Small molecule, photothermal agent		Metastatic breast cancer	[50]
	mRNA		Melanoma, HPV, anaemia, and acute lymphoblastic leukaemia and retinal problems	[51,52]
	Cyclic dinucleotide	Responsivity	Lung metastasis of melanoma and breast cancer	[53]
	mRNA, protein	Charge and responsibility	Lung and spleen diseases	[54]
	siRNA	Surface modification	Pulmonary diseases	[55]
	pDNA	Surface modification and responsivity	Osteoporosis	[56]
	siRNA	Charge and surface modification	Hepatocellular carcinoma	[57]
	NA; observed distribution	Shape	Neuroinflammatory disorders; cervical cancer	[36,58]
Polymer NPs	Small molecule	Responsivity	Non-small-cell lung cancer, lung carcinoma	[59,60]
	Protein		Diabetes	[61,62]
	Protein, small molecule		Breast cancer immunotherapies	[63,64]
	Protein, gRNA		Eye monogenetic diseases	[65]
	Protein, ssDNA		Influenza A H1N1 vaccine	[66]
	Anti-sense RNA		Mitochondrial disorders	[67]
	Cyclic dinucleotide		Pancreatic adenocarcinomas; glioblastoma	[68]
	siRNA, small molecule		Cancer	[69]
	siRNA		Metastatic melanoma	[70]
	Small molecule	Surface modification	TNBC; SLE; myocardial ischaemia reperfusion damage; breast cancer	[71]
	mRNA		Liver disease, ovarian cancer, melanoma, glioblastoma	[72,73]
	mRNA, DNA		Cystic fibrosis	[74]
	Dyes		Glioblastoma	[75]
	NA; observed distribution		Osteoarthritis; nuclear delivery	[76,77]
	Small molecule	Surface alteration and response	Ovarian, breast, and hepatocellular carcinoma cancer	[77]
	Small molecule, peptide, protein		Colorectal cancer	[78]
	siRNA, pDNA		Hepatocellular carcinoma	[79]
	Antibody, miRNA		Colorectal cancer	[80]
	Antibody, photosensitiser		Metastatic breast cancer	[81]
Inorganic NPs	Imaging agent, small molecule	Responsivity	Breast cancer	[81]
	Neoantigen, adjuvant, photosensor		Colon carcinoma, melanoma	[82]
	Photosensitiser	Surface modification	Squamous cell oral cancer	[83]
	siRNA		Breast cancer	[84]
	miRNA		TNBC	[85]
	NA; observed distribution		Neurological disorders, glioblastoma	[37,86]
	Protein, antibody	Surface modification and responsivity	Dysfunctions of mitochondria	[87]
	Small molecule		TNBC	[88]
	NPs for magnetic hyperthermia	Surface modification and size	Breast cancer	[89]
	Small molecule	Surface modification and shape	NSCC lung cancer	[90]

LNP, lipid nanoparticle; DNA, plasmid DNA; gRNA, guide RNA; NA, not applicable; miRNA, micro-RNA; siRNA, small interfering RNA; NP, nanoparticle; TNBC, triple-negative breast cancer; ssDNA, single-stranded DNA.

## 6. Biomaterial

Biomaterials have enhanced the transport and effectiveness of numerous medicinal substances, including enzymes, antibodies, vaccinations, peptides, medications, and vaccine components [91,92]. Recent developments in the fields of materials science, organic and synthetic chemistry, biotechnology, and genetic engineering have been instrumental in the development of polymer- and lipid-based materials [93] for the delivery of drugs [94]. These materials target specific body parts and release medicines over time, reducing the amount of drug needed and minimising patient side effects [95]. To optimise the therapeutic advantages, biomaterials can have their physicochemical properties and administration routes carefully modified. Biomaterials have enhanced oral and injectable drug administration [19] and created new delivery routes [96], such as pulmonary, transdermal, ocular, and nasal. Hydrogels are promising materials for wound protection. Zhao et al. determined wound healing activity in a model immersed in seawater using a chitosan-phenoxyethyl acrylate hydrogel [97] (Figure 3). Each administration route’s benefits and drawbacks must be considered when designing biomaterials for drug delivery.

The four basic systemic drug delivery methods were chosen because they increase drug accumulation on target locations and prevent long-term off-target and undesired side effects. Many biomaterials can be used as carriers or adjuvants in vaccine formulation. Biomaterials that encapsulate DNA, microRNAs, the CAR gene, and the CRISPR-Cas9 system are important gene therapy nucleic acid delivery systems. In stem cell treatment, biomaterials are used to create three-dimensional culture systems to control stem cell behaviour. Implantable scaffolds and injectable biomaterials for immune cells, medicines, and biological components are also described. For the clinical usage of biomaterials, we conclude with the issues that must be addressed. These issues include selecting the correct animal models to test biomaterial platforms and creating a simple, standardised production technique.

### 6.1. Biomaterials in Targeted Drug Delivery

Immunotherapy, molecular targeted therapy, and other recent biotherapies have greatly enhanced cancer patients’ prognosis and quality of life [98]. However, these treatments have poor tumour selectivity; therefore, there are still many questions about their safety and effectiveness when administered systemically. Even though biotherapy has strong clinical validation, only some patients show improvement after using it. Therapeutics administered in their pure form operate the risk of rapid biodegradation, unwanted biodistribution to organs and tissues that are not targets, and inadequate concentrations in diseased tissues because they are unable to penetrate solid tumour permeabilisation barriers. In addition, systemic toxicity is possible since some medications do not stay contained within the lesions [98]. Consequently, research into improving systemic medication delivery methods that are both more targeted and easier to regulate is progressing. After a medicine is administered systemically, researchers and doctors can control its pharmacokinetics with the use of biomaterials. Researchers have investigated the possibility of encapsulating or conjugating nanoparticles with a variety of medications, including cytokines, small-molecule inhibitors, immune checkpoint inhibitors, and chemotherapeutics, to facilitate the delivery and controlled release of these therapies to specific locations [98,99,100,101,102,103].

#### 6.1.1. Passive Targeting

The passive medication targeting mechanism is called improved permeability and retention (EPR) [104]. Drug distribution to on-target areas depend on carrier size, shape, surface charge, stiffness, and contact angle [105]. Research shows that macrophages absorb nanoparticles inversely proportional to their size and positively related to those larger than 100 nm [106,107]. One method involves using cationic nanocarriers and negatively charged cartilage ECM components to transfer medications to the cartilage through electrostatic interactions with positively charged poly β-amino esters (PBAEs) [108]. In general, cylindrical carriers are more easily internalised than their stiff and spherical counterparts. But these methods are not perfect, since certain medications accumulate in places they cannot, such as the liver and spleen. In addition, different absorption methods or distinct material compositions may cause different incorporation patterns to have different consequences. Additionally, there was a significant limitation to the practical translation of EPR, because its beneficial effects in animal cancer models were not observed in human cancer patients [109]. Figure 4 shows the passive targeting of nanoparticles to cancer cells.

#### 6.1.2. Nanomaterial-Induced Endothelial Leakiness

Micrometre-sized holes caused by nanoparticles with specific properties are known as nanomaterial-induced endothelial leakiness (NanoEL), a term initially used in 2013 by Setyawati and colleagues [110]. Researchers are looking at targeted DDSs to improve the safety and efficacy of long-term drug administration, by promoting effective on-target drug accumulation while minimising off-target and undesired side effects. Inorganic nanoparticles mostly composed of TiO_2_, gold, and SiO_2_ are responsible for the controllable and tuneable NanoEL effect, in contrast to the biological phenomena known as the EPR effect, which is characterised by an abnormal vascular network [111]. Damage to the intercellular junctional protein VE-cadherin and cytoskeletal modification carried on by nanoparticles are the main causes of this condition. The frequency that NanoEL occurs is inversely proportional to the nanoparticle size and endothelial cell type. An effective inducer of NanoEL are gold nanoparticles (GNPs) with a diameter ranging from 10 to 30 nm on average [112]. Endothelial cells in the human umbilical vein are less responsive to NanoEL than those in the human mammary glands and skin [112]. Because intravenous pharmaceuticals must first cross the endothelial barrier before they can reach tumour sites via the blood arteries, NanoEL enables the controlled release of nanomedicines into tumours and other areas affected by vascular diseases. But NanoEL could also create leakiness in undesired places, which could lead to metastasis, oedema, inflammation, and other unwanted effects. The researchers Peng et al. discovered that in animal models, NanoEL generated by TiO_2_, silica, and gold nanoparticles could greatly enhance cancer spread by speeding up the intravasation and extravasation of breast cancer cells [113]. To harness the promise of NanoEL for medication delivery, particularly in cancer treatment, additional research is needed to create biomaterials that are less hazardous.

#### 6.1.3. Stimuli-Responsive DDS

Both internal and external stimuli, including hypoxia, pH, ROS and enzymes, and glutathione (GSH), as well as heat, light, magnetism, ultrasound, and electricity, can trigger the release of payloads by stimulus-responsive carriers [114]. These methods improved medication administration by precisely and flexibly regulating drug release at the site of action, sparing healthy organs and tissues from exposure to the drug prematurely. DDS structural and solubility shifts or drug delivery platform conjugation cleavage in diseased tissues and microenvironments increase drug unloading. As cancer extracellular compartments or endosomes (5.85–7.68) become acidic, pH-responsive nanocarriers become unstable or fusogenic, releasing medications on target in tumours and the cytosol [115]. Specialised, easily controllable equipment is essential for the implementation of external stimuli-based DDSs. As an example, smart thermosensitive DDSs remain stable at 37 °C, which is the usual body temperature, but they release the medications they contain when the temperature rises to 40–42 °C. 

These stimuli-sensitive DDSs possess a few limitations, though. Even while infrared (IR) light is a popular external stimulus, its low tissue penetration and high phototoxicity severely limit its usefulness in many contexts [116]. On top of that, because of the high degree of variability within and between target sites, drug release is frequently facilitated by a single-responsive delivery system, which is both inadequate and sluggish. To find ways around these problems, scientists are creating materials with many responses. Chen et al. improved the vascular biocompatibility achieved by TiO_2_ nanotubes decorated with multifunctional Ag nanoparticles [117] (Figure 5). Zhong et al. developed a unique dendrimeric nanocage that could respond to changes in temperature, pH, and oxidation–reduction states; this allowed them to transport doxorubicin more effectively [27]. A nanocage was created by combining PEG derivatives with dendrimers modified with lipoic acid and then crosslinking them. At cooler temperatures, it passively accumulated at tumour sites and showed a loading capacity that was almost two times higher. In addition, specialised equipment can be used to administer external stimuli that guide the release of medications at the desired spot.

#### 6.1.4. Antibodies or Ligands Target Actively

Drugs can be actively delivered to tumour cells, immune cells, and tumour tissues through a special ligand–receptor interaction when targeting ligands are utilised. A variety of ligands, such as aptamers, peptides, and particular antibodies, fall within this category. Commonly used ligands for targeting various tumour types include monoclonal antibodies against molecules such as HER2, EGFR, TfR, CD20, PSMA, and Mam-A [118,119]. Aptamers are highly specialised macromolecules based on nucleic acids that are very adaptable. To facilitate the targeted delivery of anticancer medicines, aptamers are functionalised onto nanocarriers such as liposomes, polymeric micelles, and biomimetic nanocarriers [120]. Because of their distinct amino acid sequences, capacity to penetrate cells, and malleable conformations, peptides can also be molecularly identified. Structure-based molecular docking produces cancer-targeting peptides [121]. Polymeric nanoparticles are modified with peptides that target blood–brain barrier transferrin receptors (T7) and damaged neurons (Tet1) to improve Alzheimer’s disease drug accumulation and efficacy [122,123].

The receptors for the ligand-targeting DDS are most often studied in tumour tissues. In principle, it should be possible to target any molecule that is abnormally expressed in various tumours. As an example, Zhang et al. [124] demonstrated a method for targeting p32-positive breast cancer in vivo with a therapeutic medication by employing a conformational epitope imprinting technique on the p32 receptor and a polymeric nanoparticle smaller than 40 nm. Mice with human prostate cancer xenografts that express PSMA might exhibit an improved tumour response to a docetaxel nanoparticle that targets PSMA rather than to passively targeting nanoparticles [125]. The tumour parenchyma may not be well permeated by the therapeutic drug-carrying nanoparticles because of the vascular barrier. To address this, it is helpful to think of the chemicals expressed by endothelial cells as target receptors. Using fucoidan, a fucosylated polysaccharide with a nanomolar affinity for P-selectin. Nanoparticle encapsulation allowed for the selective delivery of doxorubicin and paclitaxel to P-selectin-positive endothelial cells, as well as many cancer cells and MEK162. In comparison to passively administered nanoparticles, this state-of-the-art DDS has the potential to promote tumour-specific infiltration, increase anticancer activity, decrease systemic toxicities, and extend local drug release. The redistribution of medicines and susceptibility could be achieved for cancers with minimal or no P-selectin expression through ionising radiation-induced P-selectin expression.

A different way to increase the diffusion rate is to target a specific group of endogenous immune cells that can access the tumour core and deliver the therapeutic medications to them. Conjugating anti-CD8a F(ab’)2 fragments to the surface of a PLGA/PEG-based antibody-modified nanoparticle was developed by Daniela et al. This nanoparticle has the potential to efficiently transport small chemicals that modulate the immune system to lymphoid tissues, tumour microenvironment regions, and the blood circulation in order to reach programmed death-1 (PD-1)^+^CD8^+^ T cells [126]. In addition to providing anti-PD-1 treatment, this nanoparticle sensitised the MC38 and B16 cancer models and increased the proportion of tumour-infiltrating CD8^+^ T cells. Contrarily, similar efficacies were not exhibited by the free agents or the non-targeting nanoparticles [126]. The most potent APCs, dendritic cells (DC), activate adaptive immunity by processing and presenting antigens to lymphocytes [127]. It is possible to customise the nanoparticle delivery of nanomedicine to DCs by modifying them to attach to certain receptors, such as the DEC205 and 33D1 receptors, which are abundant on DC surface expression [128,129,130]. Modified drug delivery has the potential to target macrophages, which are an important part of the innate immune system [131]. Nanoparticles loaded with chemicals that specifically target different kinds of immune cells might eventually make it possible to control the immune system’s activities and stop diseases in their tracks. Because they use precise ligand–receptor bindings, active targeting strategies are supposed, in theory, to have the fewest off-target consequences.

#### 6.1.5. Biomaterials in Vaccine Development

When it comes to preventing infectious diseases and cancer, vaccination is crucial. Historically, vaccines have been created using either dead or live microorganisms, or parts of microbes. Past epidemics, including polio [132] and smallpox [133] have been substantially reduced in terms of health burden due to vaccines. According to the World Health Organisation (WHO), they prevent two to three million fatalities annually caused by measles, influenza, tetanus, and pertussis. However, traditional vaccination still faces challenges, with its inferior immunogenicity and safety. Furthermore, there are currently no effective vaccines available for the immunisation of some human infections, such as the HIV virus and the malaria parasite Plasmodium falciparum [134]. The development of vaccines requires immediate innovation in response to the rising demand for high-quality vaccines. Due to its ability to shield immunological medicines from enzyme degradation and harsh pH conditions while delivering their targeted and prolonged release, a novel family of biomimetic materials has recently attracted a lot of attention as a vaccine carrier [135]. Vaccine delivery biomaterials have been covered in this article. You can obtain a more thorough overview of biomaterials used in vaccine delivery in the additional sources [136].

#### 6.1.6. Liposomes

Alec Bangham initially described liposomes in the mid-1960s [137,138]. These bi-layered vesicles can be made of either pure lipids or a combination of lipids and cholesterol. Because of their adaptability and plasticity, liposomes have long been used as adjuvants or delivery vehicles in vaccine research. According to the review of Li et al. [139], to build liposomes with specific qualities, one can change their lipid composition, surface structure, particle size and charge, manufacturing methods, and so on. The absorption efficiency of cationic liposomes by macrophages and DCs is higher than that of anionic liposomes, making them the preferred vaccine carrier type [140]. Bypassing endosomal–lysosomal degradation in cells is another capability they possess [140]. In a phase III clinical trial, patients with advanced unresectable NSCLC may have a better chance of survival if they receive a liposomal vaccination that contains the MUC1 peptide, mono-phosphoryl lipid A adjuvant, and the lipids of dimyristoyl phosphatidylglycerol (DMPG) and dipalmitoyl phosphatidylcholine (DPPC) [141,142]. To prevent rapid mRNA degradation, facilitate mRNA translation in DCs, and introduce tumour antigens to T lymphocytes, lipid nanoparticles (LNPs) are utilised. In addition, scientists discovered that particle charges, not lipid species, affected the total biodistribution of nanoparticles when they used mRNA that encoded a fluorescent protein [142,143].

#### 6.1.7. Virus-like Particles (VLPs)

Biocompatible capsid proteins, which do not contain a viral genome, self-assemble into multiprotein nanoparticles known as viral lipid particles (VLPs). Innate and adaptive immune responses can be activated by VLPs, making them great vaccine candidates because of their beneficial immunological properties, which prevent virus reproduction. The European Medicines Agency (EMA) has granted approval to MosquirixTM, the first licenced vaccination based on vector-like particles (VLPs) [144]. MosquirixTM was recommended by the EMA for 6–17-month-olds to prevent malaria. MosquirixTM requires three doses and an 18-month booster [145,146] (Table 2).

#### 6.1.8. Inorganic Particles

Inorganic nanoparticles, such as carbon nanoparticles (CNPs), silica nanoparticles (SNPs), iron oxide nanoparticles (IONPs), graphene nanoparticles (GNPs), zinc oxide nanoparticles (ZNPs), calcium phosphate, aluminium, and many more, are vital antigen carriers and vaccine adjuvants despite their poor biodegradability. Inorganic particles’ two main advantages are their solid structures and the fact that their production can be controlled. SNPs have a high drug-carrying capacity and variable porosity, and are known to be biocompatible, making them effective vaccine carriers. Antigen delivery by mesoporous SNPs can be controlled and maintained over time [147,148]. SNPs have the potential to stimulate the immunological responses mediated by T-helper 1 (Th1) and Th2 as potent vaccine adjuvants [149]. A new ciVAX vaccine was created by Mooney’s group. It consists of superparamagnetic microbeads, which capture mesoporous silica nanorods and pathogen-associated molecular patterns (PAMPs), which absorb CpG-rich oligonucleotides and granulocyte–macrophage colony-stimulating factor (GM-CSF) [150]. The effectiveness of this vaccination in preventing septic shock and infections caused by Gram-negative and Gram-positive bacteria was encouraging. Antigen transport is greatly enhanced by CNPs that possess large mesopores and macropores. They have the potential to stimulate robust IgG-, IgA-, and Th1- or Th2-mediated immune responses when used as oral vaccine adjuvants [151]. In addition, both single-walled and multiple-walled nanotubes can be produced from CNPs. As an example, small peptide antigens might be effectively delivered to dendritic cells (DCs) via single-walled carbon nanotubes, leading to robust IgG responses [152]. Carbon nanotubes have strong pro-inflammatory effects and an irregular form, although it is unclear whether these factors contribute to their in vivo toxicity. The unique adjustable and biocompatible features of GNPs are enhanced by their ease of synthesis into cores of varying sizes and shapes. Studies on influenza and foot-and-mouth disease have shown that GNPs are effective vaccine carriers [153]. To illustrate the point, GNPs were conjugated with CpG to carry an influenza A viral antigen, which in turn triggered protective immunity [154]. Inducing human T cell proliferation could be achieved by conjugating respiratory syncytial virus protein antigens with gold nanorods. Immunological vaccination adjuvants are another potential application for GNPs. Conjugated HIV vaccines with gold nanorods were able to induce humoral and cellular immunity more effectively than bare DNA vaccines [155]. Notably, GNPs have the potential to stimulate inflammatory responses and produce cytotoxicity at high dosages (>8 mg/kg) [156].

#### 6.1.9. Polymeric Particles

Numerous biocompatible and biodegradable polymeric particles, such as chitosan, dextran, polycaprolactone, PLGA, and self-assembled peptides, have been developed for the purpose of delivering vaccines [157]. Along with encasing antigens for improved delivery, these nanoparticles can biodegrade and release the antigens gradually over time [158,159,160]. The non-immunogenic deacetylated derivative of chitin, chitosan, has found widespread application in the administration of several vaccinations, such as those against DNA [161] and hepatitis B virus (HBV) [162]. Mice infected with Mycobacterium TB may be protected from infection and have an enhanced T cell response when tested with chitosan nanoparticles loaded with FL and Esat-6/3e T cell epitopes [163]. Eskandari et al. [164] highlighted the remarkable potential of self-assembling peptide molecules into targeted nanostructures for vaccine administration. Conversely, polymeric particles have the potential to function as vaccination adjuvants. For example, the insulin-derived adjuvant Advax has the potential to greatly boost the immunological response that the HBV and influenza vaccinations induce [165,166]. Because of their biodegradability and pH responsiveness, PBAEs—which were synthesised via the Michael addition of amines to acrylates—have recently attracted considerable attention as a delivery mechanism [167]. The varied compositions of PBAEs, either alone or in combination with other polymers, may fulfil various needs for targeted distribution [168].

#### 6.1.10. Outer-Membrane Vesicles (OMVs)

Gram-negative bacteria, which can vary in size from 20 to 250 nm, are the natural sources of OMVs. RNA, DNA, periplasmic proteins, enzymes, peptidoglycan, and lipopolysaccharide (LPS) are all present in OMVs’ lumen, whereas phospholipids (PL), membrane proteins, and a bi-layered lipid membrane nanostructure comprise OMVs’ lipid membrane nanostructure [169]. During their in vitro growth forty years ago, Vibrio cholerae and Neisseria meningitidis were found to form OMVs as membrane sacs [170,171]. Not only are OMVs exceedingly stable in the face of varying temperatures and treatments [172], but they can also be designed to express a wide variety of antigens [173,174]. Results from clinical trials using vaccinations based on oral microvessels (OMVs), such the Meningitis type B (MenB) OMV vaccine, have been promising. All across the globe, people could be protected from meningococcal serogroup B infections by using the multi-component MenB vaccination (4CMenB) [175]. One advantage of OMVs as vaccine carriers is their ability to express both homologous and heterologous antigens, which can be achieved by bioengineering [176].

#### 6.1.11. Immunostimulating Complexes (ISCOMs)

Hydrophobic saponin, antigens, cholesterol, and phospholipids can be mixed to spontaneously produce spherical ISCOMs, as first described by Bror Morein in 1984 [177]. ISCOMs have the potential to be engineered into stable adjuvants and vaccine carriers. For instance, following pulmonary immunisation, ISCOMs loaded with the Antigen 85 complex (Ag85) may enhance cellular and humoral immune responses to Mycobacterium TB infection [178]. Still, antigen protein inclusion was a major headache for classical ISCOM synthesisers, since it complicated the process, was not always well controlled, and limited the kinds of antigens that could be used. In response to these concerns, researchers created ISCOMATRIX [179]. This is a cage-like particle with a diameter of 40–50 nm that contains the purified portions of Quillaia saponaria extract (ISCOPREP saponin), phospholipids, and cholesterol. Because of its Quil, ISCOMATRIX is easily combined with any antigen [180]. Both the innate and adaptive immune systems are affected by this powerful immunomodulator. A variety of antibodies and antigen-specific T cell responses can be induced by ISCOMATRIX vaccines in the context of cancer immunotherapy and infectious illnesses [181,182]. Additionally, human clinical trials have assessed a range of immunogenic, safe, and well-tolerated ISCOMATRIX vaccines.

## 7. Biomaterials in Gene Therapy

Encoding DNA, microRNA, messenger RNA, small hairpin RNA (shRNA), and small interfering RNA (siRNA) are all part of gene therapy. For the treatment of many diseases, including infections, cancer, and hereditary problems, the FDA and EMA have authorised a number of gene therapies [183]. However, the absence of reliable delivery vectors has limited their widespread use. There are two main types of gene vectors: viral and non-viral. One possible explanation for the superior performance of non-viral vectors compared to viral vectors is that the former can transport bigger genetic resources without integrating into the host genome upon transfection [184]. Inorganic compounds, lipids or lipid-like substances, and polymeric materials are the three basic types of non-viral gene vectors. For therapeutic gene delivery, the most active research is presently focused on lipid (-like) and polymeric molecules. To successfully administer drugs, one approach is to encapsulate them in lipid nanoparticles (LNPs). However, LNPs administered intravenously substantially accumulate in the liver, where they are taken up by the reticuloendothelial system (RES). Nanoparticles designed to transiently occupy liver cells are called the nanoprimer (Figure 6). The pretreatment of mice with the Nanoprimer decreases the LNPs’ uptake by the RES. By accumulating rapidly in the liver cells, the Nanoprimer improves the bioavailability of the LNPs encapsulating human erythropoietin (hEPO) mRNA or factor VII (FVII) siRNA, leading respectively to more hEPO production (by 32%). The use of the nanoprimer offers a new strategy to improve the systemic delivery of RNA-based therapeutics [185,186].

### 7.1. Delivery of miRNA

The intentional modulation of gene expression patterns with foreign RNAs is possible through RNA-based gene therapy. It is necessary for the delivery vehicles to improve the stability and effectiveness of naked RNAs in RNA-based gene therapy since these molecules are quickly broken down by nucleases and then excreted by the kidneys. The designs, sizes, and intracellular locations of action of non-viral gene vectors optimised for DNA and RNA transport are distinct, making it impossible to simply adopt one for the other. To be effective, mRNA delivery vehicles should shield the cargo from enzyme destruction, elude immune detection, transport it to the intended tissue, and facilitate cell entrance. In vivo studies have demonstrated the effective administration of messenger RNA through intranasal and systemic routes by means of biomaterials, including hyper-branched PBAEs and PBAEs covered with a positively charged lipid layer [186,187,188]. With the use of these polymeric carriers, stable and concentrated mRNA polyplexes can be nanoformulated for non-invasive aerosol inhalation; this would allow for even and targeted distribution throughout the mouse lungs. Biomaterials are responsible for transporting siRNA to cytoplasmic locations and assembling it into RNAi mechanisms for its delivery to occur. The development of biomaterial-based gene therapy was greatly influenced by the 2018 FDA approval of patisiran (ONPATTROTM), the first siRNA therapy. Patisiran, an agent for peripheral nerve illness (polyneuropathy), has been encapsulated in LNPs for the purpose of targeted administration to hepatocytes [189]. Methods for delivering therapeutic mRNA and siRNA have been summarised in other in-depth reviews [190,191]. The development of miRNA delivery mechanisms was the primary emphasis of this research.

Short endogenous non-coding RNAs, called microRNAs (miRNAs), partially complement their target messenger RNAs (mRNAs) and hence adversely influence gene expression [192]. In the cytosol, pre-miRNAs are cleaved to generate mature miRNAs [192]. For miRNA transfection, synthetic polycations have attracted a lot of interest. For example, in their study on cultured primary cardiac fibroblasts and heart tissues, Li et al. [193] showed that star-like polycations based on poly (glycidyl methacrylate) were particularly effective miRNA delivery nanovectors. These vectors can be made far less hazardous by making extensive use of the secondary amine and hydroxyl groups that border them [194]. Another platinum-containing nanoscale coordination polymer was developed to enhance miRNA-655-3p transfection into colorectal cells and to extend its life in blood circulation [195]. By using this approach, tumour cell proliferation and invasion, epithelial–mesenchymal transition, and colon cancer metastasis to the liver can all be effectively halted. The clinical utility of anticancer miRNAs may be enhanced by the fact that PLGA nanoparticles could deliver them to the deep tissues of kidney and liver lesions in a pig model [196]. When correctly engineered, polymeric nanoparticles show significant promise as vectors for in vivo gene therapy miRNA delivery.

Natural materials with strong biocompatibility and polycations can be modified to facilitate miRNA distribution. A potential solution to the problem of miRNA delivery to cardiac macrophages following a myocardial infarction is to combine miRNA with hyaluronan and sulphate to form slightly anionic nanoparticles [197]. By mediating cardiac-targeted delivery and effectively modulating the inflammatory and reparative state of lesions, hyaluronan-sulphate can enhance angiogenesis and left ventricular remodelling. In vitro studies using hepatocellular carcinoma (HCC) cells showed that peptide-based vectors may boost miR199a-3p levels by more than 500-fold, promote their accumulation at tumour sites, down-regulate the mTOR gene, and reduce tumour development by more than half [198]. To facilitate targeted delivery, these peptides can self-assemble into nanoparticles that can be coupled with targeting ligands. Exosomes are the tiniest vesicles outside of cells that have several desirable characteristics, such as being biocompatible, having low toxicity, having a deep penetrating property, etc., and therefore being an ideal vehicle for tiny RNAs, such as siRNA and miRNA [199]. For example, a mouse model was able to effectively suppress mRNA and ameliorate CCL4-induced liver injury with miR-155 given by functionalised exosomes [200]. The distinct benefits of natural material-based delivery methods, such as their biodegradability, intrinsic biocompatibility, ease of synthesis, reduced cost, and variable designs, have attracted considerable interest in miRNA delivery.

Notably, biomaterials can be altered to utilise their different activities by co-delivering multiple therapeutic nucleic acids at the same time. By utilising nanocarriers made of PEI and dexamethasone, Kim et al. were able to co-deliver pDNA and shRNA to human mesenchymal stem cells (MSC). This allowed them to over-express or knock down proteins linked with differentiation, leading to morphological alterations in MSC [201]. Furthermore, targeted gene silencing and the simultaneous targeting of numerous disease-related pathways are both possible outcomes of the therapeutic delivery of miRNA and siRNA in combination. To delivering miR-200c and siPlk1 siRNA simultaneously, multifunctional tumour-penetrating mesoporous silica nanoparticles were created [202]. A photosensitiser called indocyanine green was used for endosomal escape, whereas an iRGD peptide was used for deeper tumour penetration. Both the main tumour’s development and its metastasis were significantly reduced by this method, which demonstrated a combination cell-killing activity. Ultimately, biomaterials can greatly enhance the curative potential of RNA-based treatments. Research into optimising various biomaterials for the delivery of intact genetic resources to their target region (cytosol/nucleus) is underway in both in vitro and in vivo settings. This includes lipid-based materials and polymeric nanoparticles. There are still certain challenges that must be overcome. Two of the most widespread worries are the possibility of toxicity and the occurrence of immune responses that are unintended in relation to biomaterials [183].

### 7.2. Delivery of Encoding DNA

As a “knock-in” method, delivering encoding DNA to the target cells allows them to express therapeutic proteins. Inorganic nanoparticles, polymers, lipid-based materials, and peptide-based materials are among the DNA-based synthetic delivery agents now under development for gene delivery [203]. Particularly in cancer trials, several lipid-based and polymeric DNA vectors have progressed to the clinical evaluation phase [203]. Lipid nanoparticles can bind and compress DNA due to their cationic and ionisable headgroups. Additionally, cationic lipids mediate electrostatic interactions between negatively charged nucleic acid or plasma membrane components, allowing for effective DNA delivery and cellular uptake. Raising the lipid membrane’s fluidity improves transfection efficiency [204]. By utilising liposome–protamine–DNA nanoparticles to transport pDNA, Rajala et al. developed a synthetic virus that improved gene deficiency-related issues in vivo and allowed for more efficient and sustained gene expression [205]. Despite their widespread usage as transfection reagents, PEI and its derivatives are harmful due to their non-degradable nature [206,207,208]. In comparison to the widely used PEI, SuperFect, and Lip-ofectamine, the most advanced gene transduction agents currently available are the highly branched PBAEs, which offer less toxicity and a significantly greater transfection efficacy. Furthermore, even after a year of freeze storage, DNA encapsulated with PBAEs retains its complete transfection capability (Figure 7).

Due to their three-dimensional shape and many terminal groups, dendritic or branching biomaterials outperform their linear counterparts as gene vectors. The transfection efficiency might be increased by 8521 times using highly branched PBAEs compared to their linear counterparts, according to research by Zhou et al. [209]. Potentially useful for treating inherited skin illnesses, PBAEs that blend linear and branching designs have been found to efficiently transfer minicircle DNA into fibroblast cells, which are notoriously difficult to transfect [210]. Thus far, pDNA and other forms of double-stranded DNA (dsDNA) have formed the basis of the majority of nonviral gene vectors. To improve tissue penetration, researchers have concentrated on creating smaller nanocarriers made of single-stranded DNA (ssDNA) [211]. This delivery approach has the potential to promote transgenic expression in tumour cells, ultimately generate a major anticancer effect, and break through the thick stromal barriers of pancreatic tumour cell nests by systemic injection. For this reason, biomaterials have shown promising results in in vitro, in vivo, and clinical investigations for delivering encoding DNA. It is possible to modify their structures such that they efficiently express genes despite obstacles at the cellular, tissue, and intracellular levels. Additionally, biomaterial–DNA complexes are stable, which simplifies their manufacturing and storage conditions, increases the duration of gene expression, and favours the bench-to-bed translation. However, lipofectamine and PEI, two of the most used gene transfection vectors, are extremely hazardous, which helps to explain the reason they have not found widespread use in clinical settings. To make non-viral gene vectors safer without reducing their efficacy, a lot is still needed.

### 7.3. Therapeutic Drug–Nucleic Acid Co-Delivery

Research has also examined multimodal administration approaches to improve therapeutic efficacy and lessen side effects. Lin et al. developed nano pro-drug co-delivery systems for cisplatin and autophagy inhibiting Beclin1siRNA using cationic peptides. PEGylated DSPE-PEG improved stability and biocompatibility, with cRGD delivered to cancer cells. After intravenous injection, this co-delivery approach outperformed free cisplatin treatment against cisplatin-resistant tumours and overcame acquired resistance [99]. In a similar vein, seeded retinoblastoma was treated with LNPs co-delivering melphalan and miR-181a [212]. In a rat model of a retinoblastoma xenograft, these two medicines worked synergistically to reduce the viability of cultivated retinoblastoma cells and increase the therapeutic efficacy compared to monotherapy.

### 7.4. Delivery of CAR Gene

Hematologic cancers have responded very well to CAR-T cell therapy, and solid tumours have shown encouraging signs of improvement [213]. Genetically engineered T cells called CAR-T cells permanently display the cancer-targeting CAR on their surface, enabling them to target and destroy tumour cells. There has been a lot of recent interest in using cationic polymers and LNPs to deliver the CAR gene to various lymphocytes. Yu et al. [213] produced a PEI-PAMAM self-assembling nanoparticle to transiently transfect Jurkat cells with a plasmid vector carrying the EGFRvIII CAR gene. It is possible to drive T lymphocytes to target and eliminate EGFRvIII-positive cancer cells by expressing EGFRvIII-CAR on cell membranes. Ionisable LNPs have been utilised in CAR-T cell ex vivo engineering based on messenger RNA, and they have demonstrated cancer-killing efficacy on a par with CAR-T cells produced via electroporation or viral vectors [214]. Notable among effector lymphocyte subtypes are natural killer (NK) cells. One promising new approach to cancer treatment is CAR-NK cells. To transport pDNA containing the EGFR-CAR gene into NK cells, Kim et al. created multifunctional nanoparticles [215]. These nanoparticles were made with a core–shell that was coupled with cationic polymers and a coating layer that was coated with polydopamine. Enhanced EGFR-CAR expression and cytotoxicity against human breast cancer was observed in the modified NK cells [215]. Clinical practice has refined the present methods for producing CAR-T cells, which include T cell isolation, ex vivo activation, proliferation, genetic modification, and reinfusion. Because of this, many patients cannot afford CAR-T therapy. The two commercially available CAR-T products, Kymriah^®^ and Yes-carta^®^, have yearly course costs of USD 475,000 and USD 373,000, respectively [216]. In situ T cell programming with nanoparticles is a possibility. In 2017, Smith et al. introduced CD19-specific CAR genes into circulating T cells in situ using polymeric nanoparticles with pDNA, PBAE 447, nuclear localisation peptide, anti-CD3ε f(ab)2 fragments, and poly-glutamic acid antibody [217]. These reprogrammed T cells may induce a long-term remission in animals with leukaemia, similar to infusing CAR-T cells transduced ex vivo with viral vectors. An injectable polymeric nanocarrier transcribed with messenger RNA transiently expressed CD19-specific CAR and HB core-specific TCR in circulating T cells. In situ T cell reprogramming would simplify clinical techniques, and repeated mRNA nanodrug administration can promote antitumour responses in mice against prostate cancer, lymphoma, and HBV-induced hepatocellular carcinoma [218]. This nanomedicine may be clinically useful because of its high efficiency, quick scale-up, low cost, and customisability. Nanoparticle-based CAR-T cell engineering may be limited by recurrent dosage and multi-component manufacturing, which may create safety concerns.

## 8. Biomaterials in Stem Cell Therapy

Undifferentiated cells, called stem cells, have the remarkable ability to both self-renew and develop into the specialised cells needed by different parts of the body. There are other biological uses for stem cells, including tissue engineering and regenerative medicine, which have led to the increased interest in stem cell therapy [219]. Still, in vivo transplanted stem cells frequently differ in appearance from those seeded in conventional 2D preparations. This is due to the fact that the endogenous 3D microenvironment cannot be well represented in the 2D, rigid environment [220]. However, the physiological relevance of 3D culture methods is enhanced, since they supply cells with essential chemical and physical supports. More cells are produced by 3D culture systems compared to 2D cultures with the same amount of space, and these systems are naturally more scalable. Various natural materials have been utilised to produce three-dimensional cell culture scaffolds, including collagen, alginate, gelatin, laminin, and decellularised extracellular matrix (ECM). They have not been extensively studied or used in clinical settings due to the difficulties of biochemical alteration and the risk of immunological rejection [221]. Three-dimensional scaffolds made of synthetic materials, such organic and inorganic porous materials, can be biochemically functionalised, mechanically adjusted, or surface-changed to resemble the ECM found in nature. For the non-destructive imaging of cell function, polymer-based hydrogels have potential because of their ECM-like characteristics and optical clarity [222]. Because of their ability to mimic the microenvironment, hydrogels are commonly used for injectable biomaterial-based stem cell transplantation. To make them, oligomer precursors are usually crosslinked, either chemically or physically (Figure 8).

Specialist microenvironments with well-defined extracellular factors are necessary for stem cell proliferation, differentiation, attachment, and migration. Reproducible cell multiplication for transplantation, the preservation of stemness, and prevention of cell death are key challenges to the success of stem cell therapies at present. It is possible to manipulate stem cell behaviours and functions using biomaterials by enhancing these variables. Bioactive motifs and cell-binding domains can be chemically conjugated or adsorbed onto synthetic biomaterials to form the matrix. For example, by activating the downstream kinase signalling cascade, biomaterials engineered to bind with integrins could enhance cell adhesion and create induced pluripotency stem cells (iPSCs) [223]. Through HA-CD44v6 interaction, hydrogels based on HA may enable the invasion of cells with high levels of the HA receptor CD44v6. Stem cell behaviour is significantly affected by the elasticity of non-porous and porous materials. While several non-degradable hydrogels mimicked the natural ECM in certain physiological ways, they were mostly just elastic, in contrast to the viscoelastic ECM [224,225].

One of the most important factors in regulating the behaviour of various stem cell populations in a three-dimensional environment is the stiffness of the matrix. Human mesenchymal stem cells (MSCs) can be easily redirected from adipogenesis to osteogenesis by increasing the rate at which stress stiffening occurs in soft responsive hydrogels derived from synthetic polymers. The cell’s shape is maintained and regulated by the stress stiffening-dependent microtubule cytoskeleton, which undergoes deformation and reorganisation in conjunction with this action [226]. It should be mentioned that the stemness preservation of low contractible neural progenitor cells (NPCs) was unaffected by changing the initial hydrogel stiffness (~0.5–50 kPa) [227]. Thus, it was hypothesised that only the types of stem cells with a high contractile capacity were affected by the matrix stiffness. Hydrogels’ ability to relieve stresses and reconfigure their matrix in response to external stimuli means they can control stem cell destiny and activity. In a two-dimensional cell culture setting, cells can freely spread out; but, in a three-dimensional hydrogel, cells can only migrate and proliferate after the matrix has been remodelled [227].

As a result, research into creating an adaptive biomaterial that can change over time to influence cellular fate and function is gaining momentum. To better understand how cells interact with their dynamic mechanical environment, an ECM that can be mechanically switched on and off in reaction to chemical and physical stimuli is ideal. This is crucial for regulating stem cell proliferation and differentiation in regenerative medicine and tissue engineering, and for comprehending physio-pathological processes. The reversible patterning of cell-laden gel biomaterials through targeted modification with bioactive site-specific proteins has recently been achieved using a method known as 4D patterning. This approach has the potential to pave the way for the first-of-its-kind spatial and temporal control of complex biological processes [228]. Furthermore, the development of tunable materials was influenced by the reciprocal interactions of the ECM and stem cells. Dynamically controlling the behaviours and fate of MSCs was the goal of Jia et al. [229], who constructed adaptable liquid interfacially formed protein nanosheets. Here, stem cell traction force may cause ECM protein spatial rearrangement, which disrupts cell destiny.

### Stem Cell Carriers

The effectiveness of stem cell therapy is highly dependent on the cells’ ability to survive and function both during and after transplantation. Stem cells can be delivered and protected from mechanical pressures using biomaterials, which can also create a microenvironment that supports their optimal function at a given place. In addition to cells, these materials have the potential to enhance the function of transplanted stem cells by co-delivering therapeutic medicines such as anti-inflammatory medications [230]. Injectable hydrogels and extremely effective stem cell patches are the recent highlights of biomaterial-based stem cell delivery techniques. When it comes to cell distribution for tissue engineering, biomaterial-based stem cell patches may provide advantages that surpass the convenience of injectable hydrogels. Microsurgical instruments can be used to implant these patches, which provide strong mechanical, contractile, and electrical characteristics to aid the transplanted stem cells in their intended location [231]. Compared to injectable hydrogels, epicardial patches were able to boost cell survival fivefold in a study of biomaterials for MSC distribution in rat models of myocardial infarction [232,233,234]. These investigations may be useful for directing the practical translation of stem cell patches by evaluating their therapeutic effectiveness and safety in large animal models of ischaemia cardiac damage. Ocular disorders are another potential use for stem cell patches. In models of age-related macular degeneration (AMD) in rats and pigs, patches of human retinal pigment epithelium (RPE), grown from induced pluripotent stem cells (iPSCs) embedded in biodegradable scaffolds, improved RPE integration and function, reversing retinal degeneration [235]. In two cases of severe AMD-related visual loss, RPE patches made from ESCs were able to stabilise and improve eyesight for a minimum of twelve months [236].

## 9. NPs for Cancer Therapy

Worldwide, cancer continues to rank as the second most deadly disease [237]. The development of successful cancer medicines is incredibly tough, in part due to the complexity of cancer and its heterogeneity. The development of specific chemotherapeutics that can treat patients who display specific biomarkers has, however, led to precision medicine’s emergence as a promising method. Imatinib’s clearance by the FDA paved the way for an array of other effective targeted chemotherapeutics [238,239,240]. But better delivery could make these and other medicines even more effective. Imatinib, when administered via an NP system, improved tumour accumulation and regression in vivo, leading to a 40% survival rate after 60 days in a mouse model of melanoma [241]. This includes small-molecule medicines that have a limited water solubility and antibodies that have low stability [242]. In a similar vein, the effectiveness of many chemotherapeutics is limited due to off-target toxicity and adaptive resistance. On top of that, the tumour location is one of many cancer-related biological barriers. An improvement in distribution methods might reduce a lot of these concerns. Personalised therapies and delivery systems allow us to make the most of our understanding of cancer patients and their unique treatments.

### 9.1. Active Targeting to Cancer Cells

Numerous mechanisms and locations of action are present in existing chemotherapeutics. Doxorubicin and platinum medicines are examples of those that damage DNA in the nucleus; other drugs operate in the cytoplasm or impact organelles like mitochondria [243]. There is a requirement for NP trafficking to these areas since therapeutic doses of each medicine must be given for them to work correctly.

The specific and effective uptake of NPs can be induced by ligands such as sugars, antibodies, and others on the surface of NPs. Antibodies, transferrin, integrin ligands, peptides, glucose, [244], and folic acid are some examples of cancer cell-targeting moieties (Figure 9). Some systems now use a single NP with many targeting modalities, due to these evolving technologies. Some of these targeting strategies, like folic acid, are applicable to a wide range of malignancies; however, to determine whether a receptor or ligand is overexpressed, tumour profiling is typically necessary [245]. To clarify, not every receptor-targeting strategy enhances specificity. Receptors that are overexpressed in cancer cells can also be found in healthy tissues, which can reduce their effectiveness. Figure 9 illustrates a strategy to selectively target cancer stem cells (CSCs) using nanoparticles to deliver therapeutic agents. Differentiated tumour cells are mature, specialised cells that make up the bulk of a tumour. The nanoparticle is engineered with a targeting moiety that specifically binds to a 1. The cancer stem cell (CSC) marker ensures that the nanoparticle may preferentially attach to cancer stem cells rather than to differentiated tumour cells. The targeting moiety on the nanoparticle recognises and binds to the CSC marker on the surface of cancer stem cells. This interaction is mediated through the linker chain, which maintains the proper orientation and function of the targeting moiety. Once the nanoparticle is bound to the CSC, it releases the chemotherapeutic agent directly into the cancer stem cell. This targeted delivery helps to kill CSCs more effectively, while potentially minimising damage to healthy cells and differentiated tumour cells. By specifically targeting CSCs, the nanoparticle-based approach aims to eliminate the most treatment-resistant cells in the tumour, potentially reducing the likelihood of relapse and improving overall treatment efficacy.

The balance between cellular absorption and residence time in the circulation is another common trade-off. Detachable stealth corona systems and charge reversal systems have been built into NPs recently to optimise both features [246,247]. These systems use negative or neutral charges for circulation and positive charges for absorption. Using an MMP-degradable linker, one technique attaches PEG to the surface of the NP. When this system is exposed to the tumour microenvironment, the coating is broken down, revealing a peptide that can penetrate cells [60]. This allows for the creation of systems that can adapt a specific property to overcome the delivery obstacle they are now experiencing.

When considering active targeting in drug delivery, it is crucial to focus on transcytosable nanomedicine, which refers to nanomedicines designed to cross blood vessel walls (vasculature) to reach their target tissues [248]. This is an essential aspect of drug delivery because effective treatment often depends not only on targeting specific cells within the tissue but also on the ability of these nanomedicines to penetrate the blood vessels to access those cells. The concept of transcytosis—the process by which substances are transported across the endothelial cells of blood vessels—was notably proposed by Dr. Dvorak in the early development of drug delivery systems. Dvorak’s work suggested that targeting the mechanism of transcytosis could significantly enhance the delivery of drugs to tumours, where the leaky blood vessels often provide an opportunity for nanomedicines to enter. This approach underscores the importance of designing nanomedicines with the capability of overcoming vascular barriers, ensuring they reach their intended therapeutic targets more effectively [249].

### 9.2. NPs for Immunotherapy

Immune checkpoint inhibitors have the potential to improve cancer treatment [248]; however, there are still issues with immunomodulators’ efficacy, patient variability, and off-target effects [249]. Because proteins and other immunotherapeutics have a low delivery potential if delivered, NPs may greatly increase the delivery by shielding these drugs and making them more effective at interacting with immune cells [250].

### 9.3. Immune Activation

The immune system is designed to kill cancer cells, yet some genetically predisposed cancer cells may evade or suppress immune cells. Cancer vaccines resensitise cells with patient or allogenic tumour antigens, to teach the immune system to recognise cancerous cells. Sipuleucel-T, an FDA-approved cancer vaccine, uses tumour-specific recombinant antigens but has limited efficacy. Other research groups have generated synthetic peptides and tumour lysates for patient customisation [251,252,253], although these drugs have not yet reached clinics. NPs preserve these antigens, deliver them to the target immune cells, and reduce off-target effects. APCs that take up these NP systems prime and activate T lymphocytes with antigen [49,254,255,256]. Plastic (PLGAs), lipid-based (liposomes, LNPs), inorganic (gold, silica), or biologically generated (cell-membrane vesicles) options are used in these systems. Clinical trials of NP-derived cancer vaccines are underway [148,257,258,259]. Late-stage clinical trials of NP vaccines against COVID-19-causing SARS-CoV-2 were successful. COVID-19 antigen-coding mRNA is encapsulated by LNPs by BioNTech and Moderna. On November 30, 2020, Moderna and BioNTech/Pfizer submitted it for Emergency Use Authorisation after meeting phase III efficacy end targets. In this and other applications, the NP architecture, material properties, and active targeting alter the cellular uptake, antigen presentation, and immune response [260]. 

NPs increase immune activation specificity by targeting macrophages, B cells, and dendritic cells. Passive targeting optimises particle size and shape ratios and engages negatively charged cell membranes with positively charged particles [261,262]. Endocytosis by APCs’ numerous carbohydrate-recognising lectin receptors is cell-specific active targeting. C-type lectin domain family 9 member A (CLEC9A) and lymphocyte antigen 75 (DEC-205) can target dendritic cells on some APCs [263]. Dendritic cells, macrophages, and tumour-associated macrophages may be targeted by mannose [72,88,264], which also targets dendritic cells [265]. Macrophages can access galactose-, dextran-, or sialoadhesin-coated particles [266]. Lipoprotein-surfaced NPs activate dendritic cell SRB1 receptors, and CD19-targeted NPs actively target B cells [267,268]. Modified NPs can collect at immunological antigen-producing organs such as the spleen and liver [269]. Polymeric hydrogels and scaffolds that recruit immune cells may improve APC interactions. These systems recruit and retrain APCs with APC-targeted NPs [270]. All of these approaches aim to increase antigen–APC interaction to improve antigen-based treatments and lower dosages.

Cytosolic double-stranded DNA, which is usually found in infections, can activate the stimulator of interferon genes (STING) pathway, which in turn activates immune cells and has antitumour effects. The anticancer effects of STING agonists, which are usually cyclic dinucleotides, are encouraging; however, their instability and strong polarity hinder their absorption by cells [271]. One formulation of STING NPs boosted survival for at least 80 days in mice, and NPs improve the delivery of STING agonists [53,272,273]. Furthermore, independent of their payload, certain NPs can trigger STING via their cyclic structures, which resemble double-stranded DNA (cyclic lipids).

There are alternative methods of immunotherapy that aim squarely at T cells. Targeting T cells with NPs has been accomplished using a plethora of targeted strategies. Nuclear particles (NPs) that target PD1 [126], CD3, and THY1 (CD90) are some examples. Regulatory T cells, a subtype of immunosuppressive T cells, have been targeted using the tLyp1 peptide, which is normally employed for tumour targeting. Monoclonal antibodies that target PD1, PDL1, or CTLA4 are commonly used as checkpoint inhibitors, an immunotherapy technique that boosts the immune system against cancer. Stability considerations restrict the use of free antibodies, as they do in other contexts. Further, a strong response is observed in fewer than one-third of individuals treated with these checkpoint inhibitors. Aiming to enhance the efficacy and decrease the adverse effects of these medicines, NPs have been developed for the delivery of monoclonal antibodies (anti-PD1) [274,275], and other NP formulations introduce siRNAs, which interfere with immunological checkpoints [276]. Metastatic and blood malignancies have also demonstrated potential when treated with genetically engineered T cells. To enable T cells to target and destroy malignant cells selectively, they are engineered to express transgenic T cell receptors (TCRs) or CARs [277].It is already possible to harvest T cells from patients and expand them in vitro with the use of artificial APCs; however, novel NP formulations may make it possible to translate this approach to an in vivo setting [278]. Like conventional NPs, artificial APCs can affect T cell activation depending on their design, materials, and targeting [279]. Possible alternatives to NPs for CAR T creation include delivering CAR-encoding DNA in vivo and CAR-encoding mRNA to generate transiently changed T cells, both of which simplify antigen delivery to T cells [214,217].

### 9.4. Immune Suppression

Another consequence of improper immune modulation, known as hyperactivation, is the development of diseases like rheumatoid arthritis and systemic lupus erythematosus. T cells and B cells become hypersensitive to self-antigens in autoimmune disorders [280]. Commonly prescribed to patients with autoimmune illnesses are general immunosuppressants, which carry the risk of major adverse effects. Improved immunotherapy targeting conditions brought on by immunological overactivity is a promising area of research.

Some of the cells that can be suppressed by the immune system are APCs [281], autoreactive T cells and B cells [282], and regulatory T cells and B cells [283,284]. A tolerance to antigens, reduced reactive cells, or the reprogramming of cells are all goals of antigen-specific immunotherapy. One possible way to modify the immune system without lowering systemic immunity is through antigen-specific immunotherapy, which targets a specific subgroup of immune cells. Immune suppression makes use of active and passive targeting strategies that are comparable to those employed in immune-activating treatments. Coated NPs containing anti-CD2/CD4 antibodies, for instance, can increase the number of regulatory T cells in circulation by targeting T cells, while non-coated NPs at the same dosages could not achieve this. Sglecs, which are immunoglobulin-like lectins that bind sialic acid, can also be utilised to target B cells, and promote tolerance. The administration of immunosuppressant drugs is another method for inducing immune tolerance. Since the systemic administration of active vitamin D3 might lead to hypercalcaemia, NP delivery offers a potential alternative method. The delivery of immunomodulators and the prevention of allograft rejection have both made substantial use of PLGA NPs [285]. It is possible to administer the popular immunosuppressant tacrolimus locally and continuously for 28 days using PLGA NPs attached to a hydrogel [286]. Reprogramming immune cells at the genomic level by genetic engineering may have longer-lasting impacts [287].

### 9.5. Tumour Microenvironment Adaptation

Because it impacts the efficacy of chemotherapy, the tumour microenvironment has a significant impact on patient prognosis [287]. The EPR effect and FDA approval of early NP systems give NP-based delivery optimism; however, these systems do not enhance patient survival. Smart NP designs need to improve cargo transport and microenvironments before they can enhance therapy. To enhance their accumulation in malignant tissue, for instance, NPs can have cell membranes attached to them. Patient-derived cancer cell lines cling to NPs encased in membranes taken from the patient’s own cancer cells; however, poor targeting occurs due to a donor–host mismatch [71,288]. NPs encased in leukocyte or macrophage membranes can identify tumours, and hybrid membranes, like erythrocyte–cancer cell hybrids, can enhance the specificity even more [289,290,291]. When compared to the free drug, the activity of NPs that incorporate these membranes is two to three times higher. Similarly, NPs can be made to diffuse preferentially to specific tissues based on their material features. As an illustration, a conjugate of a poly (β-amino-ester) (PBAE) terpolymer and a PEG lipid was fine-tuned for localisation in the lungs, and it outperformed the pre-optimised form by a factor of two in both laboratory and living organism tests. We have produced additional PBAE polymers that selectively target glioma cells rather than healthy cells in vitro [292].

### 9.6. Bacteria as a Transport Vehicle

There has been a major influence on biomedical research from the continuous development of nanomaterials for different ailments. Unfortunately, nanoparticles are not very effective against any one disease because their drug delivery capabilities are not strong enough to reach the affected area. There are several benefits of using microbes in conjugation with nanomaterials rather than chemical synthesis methods. These include biocompatibility, low toxicity, ease of synthesis, and low cost [293]. Some varieties of bacteria have flagella that allow them to move about on their own. Nanoparticles (NPs) can be easily attached to living systems like bacteria or cells through chemical reactions called bio conjugation. The outcome is the use of bacteria as a vehicle for nanomaterial delivery, which facilitates drug release by penetrating more easily.

For therapeutic drugs or NPs to effectively impact the entire tumour cell population, they must be dispersed uniformly across the tumour. However, the perivascular regions of tumour edges accumulate particles, and this scenario does not happen very often [294]. This causes tumours to grow in areas where the medication concentration is low, which encourages the proliferation of dormant cells that are mostly immune to chemotherapy. Antitumoural therapy must thus specifically target the hypoxic and deeper parts of the tumour if it is to achieve its full therapeutic potential. The ability of bacteria to propel themselves and guide themselves makes bacterium-mediated tumour therapy (BMTT) ideal for intratumoural targeting [295]. They may be able to move away from the tumour’s blood vessels and get to deeper places where the conditions are better for bacterial growth. Regardless of hydrodynamic constraints, flagellated bacteria can penetrate tissues due to their intrinsic motility [296]. In the time after BMTT became a popular method for effectively penetrating and fighting tumours, numerous bio-hybrid nanocarriers composed of various nanomaterials were found. Therapeutic medications, genes, or proteins have been transported to deep tumour areas inaccessible to traditional chemotherapy using these nanocarriers. The usage of nanoparticles, however, affects the outcomes.

In the case of inhalation damage, nanoparticles mediated by bacteria can also successfully cross the sputum barrier. The thick, viscous sputum barrier is formed by airway burning or inhalation injury; it causes respiratory obstruction and serious lung infection. A crucial obstacle to effective medication administration in this instance is the nasal mucus gel layer [297]. Nanomaterials mediated by sputum-penetrable bacteria have great promise for resolving these concerns through the application of nanotechnology. To combat the issue of NP penetration, it may be possible to introduce genetically modified facultative anaerobic bacteria like Salmonella or E. coli. Together with new clinical strategies, the vast therapeutic toolbox that GMOs provide has the potential to revolutionise illness treatments, making them safer and more effective (Table 3).

### 9.7. Carrier-Free Nanomedicine Delivery Systems

There has been an inability to accumulate therapeutically effective dosages of drugs due to nanomedicine carriers’ fundamentally low drug-carrying capacity (about 10% wt/*v*). Here, carrier-free nanoagents have come a long way because of their “all-in-one” platform functionality, high drug-loading capacity, and ease of production. However, the accuracy of their distribution is limited by inherent flaws. Serious adverse effects, impaired therapeutic efficacy, and increased development risks can result from over-treating chemicals during the preparation period [298].

There have been a lot of attempts to improve the transport and release behaviour of carrier-free nanodrugs by perfecting their physicochemical properties. Their therapeutic efficacy can be significantly impacted by the in vivo complex milieu and different biological obstacles, according to studies [299]. By fine-tuning the drug-to-agent ratio in carrier-free nanoagents, we can manage the size, shape, and surface charge of the NPs, which greatly affects their in vivo fate, including the duration they stay in circulation and how much drug they release. One approach to these problems is to modify the surface of the NPs with various polymers, small molecules, or proteins, such that they are more stable without compromising their ability to carry drugs. While the existing methods can alleviate some of the concerns about early medication release, further research is required to resolve the most pressing concerns.

## 10. Prospects and Advancement

Research into biomaterials-assisted biotherapies has grown rapidly in recent decades, stimulating advancements in related interdisciplinary fields such as nanotechnology, molecular biology, and material science. A number of products have either made it to market (Table 4) or are undergoing human clinical trials, due to biomaterial-assisted biotherapies enhanced therapeutic efficacy and reduced adverse effects. Doxil^®^, the first nanodrug to receive FDA approval, passively targets tumour areas by encapsulating doxorubicin in PEGylated liposomes and keeping them in circulation for an extended period [300]. Doxil^®^ outperforms traditional doxorubicin in terms of antitumour activity across a wide range of neoplastic illnesses, while significantly reducing cardiac toxicity [301]. LNPs outperform other biomaterials in terms of loading capacity and tolerability, making them promising delivery vectors for genetic materials. Among the polymeric products that have made it to market are hormone therapeutics based on PLGA microspheres. Since VLPs are structurally and functionally very similar to real viruses, they greatly enhance the immunogenicity of vaccines based on VLPs, making them attractive vaccine carriers. Commercialised VLP vaccinations such as Cervarix^®^, Gardasil^®^, and Gardasil9^®^ have been found to induce a protective immunity against HPV infection for life [302,303]. Furthermore, clinical trials are currently underway for several more sophisticated biomaterials. One example is CRLX101, a nanocarrier made of pH-responsive cyclodextrin that contains camptothecin [304]. It has finished phase II clinical trials (NCT01612546). VLP-based HPV vaccination (Cervarix^®^, Gardasil^®^, and Gardasil9^®^) is one example of a biomaterial-assisted vaccine that has achieved commercial success. However, biomaterial-assisted biotherapies continue to encounter an array of obstacles. Before anything else, it is crucial to minimise the translational barriers to human applications by carefully selecting the animal models to evaluate these platforms. The issue of enhanced permeability and retention (EPR) has been discussed earlier by many researchers. EPR refers to a phenomenon where nanoparticles or macromolecules accumulate in tumour tissues more than in normal tissues due to the leaky blood vessels in tumours and poor lymphatic drainage. The EPR effect has been widely explored in scientific research, especially in the context of drug delivery systems for cancer therapy. The author acknowledges that the topic is intricate and multifaceted, involving various factors and variables that make it challenging to fully understand or resolve. Understanding the EPR effect adds an additional dimension to the discussion of drug delivery systems or related topics.

Animal models that closely mimic human diseases should be used to study these synthetic biomaterials [305]. Hence, additional investigation into the pros and cons of various animal models, as well as the variations in immune responses between people and these animals, is required [306]. The response to biotherapies in preclinical models may be impacted by characteristics such as housing, food, sex, and commensal microbiota [307]. Second, to ensure low batch-to-batch variability, it is recommended to use basic and standardised fabrication techniques. During the initial phases of biomaterial design, it is crucial to thoroughly examine numerous aspects that are crucial for clinical translation, such as scalability, complexity, production cost, and stability [308]. In addition, platforms that use materials that have been approved by the FDA may streamline regulatory processes and have a better chance of being used in clinical operations. Thirdly, while biomimetic nano-minerals have shown biocompatibility in animal models, the biosafety of most biomaterials has not been evaluated in humans for therapeutic use. Systemic toxicity, long-term adverse effects, long-term efficacy, and pharmacodynamics are some of the remaining challenges. For preclinical studies to be considered, they must include a big enough sample size and sufficient time for follow-up, and they must also include tests on larger mammals, such as dogs and non-human primates.

More and more people have begun to pay attention to multimodal therapies, which integrate biomaterial-assisted biotherapy with other forms of radiation, chemotherapy, or photothermal therapy [309]. Improvements in intelligent material design enabled by big data and artificial intelligence (such as machine learning) are also anticipated to hasten the development of biomaterial-based biotherapy [310]. Additionally, high-throughput screening and preclinical investigations may be made easier by combining computer technology with organoids, genomics, and proteomics [311]. However, not enough is known about the dose–effect relationship in biomaterial-based biotherapy. Research into the dosing window for safer and more effective delivery methods should be prioritised in future studies [312]. Another area of innovation in precision medicine is biotheranostics, which involves the use of biomaterials in both diagnosis and treatment.

## 11. Conclusions

To address the diverse biological obstacles encountered by different patient groups and diseases, this review has covered a wide range of NP designs that have been optimised for therapeutic administration. Patient comorbidities, different disease development stages, and individual physiologies amplify these obstacles to delivery. Various NPs developed for certain patient populations, diseases, or combinations of the two can address this wide range of requirements. Size, charge, shape, responsiveness, and surface characteristics are just a few of the NP platforms’ changeable attributes that can be chosen to optimise delivery for a particular application, treatment, and patient population. Together, precision medicine therapies and this level of personalisation can improve the patient stratification methods used to screen NP platforms, increase the number of people who can access precision therapeutics by rendering them available to more people through improved delivery mechanisms, and boost the overall effectiveness of the treatments.

## Figures and Tables

**Figure 1 pharmaceutics-16-01076-f001:**
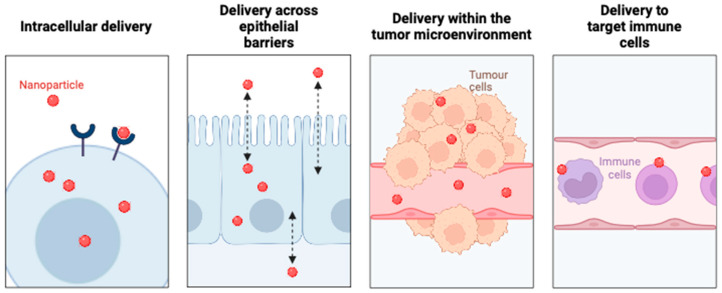
Schematic representation of biological barriers that nanoparticles can help overcome.

**Figure 2 pharmaceutics-16-01076-f002:**
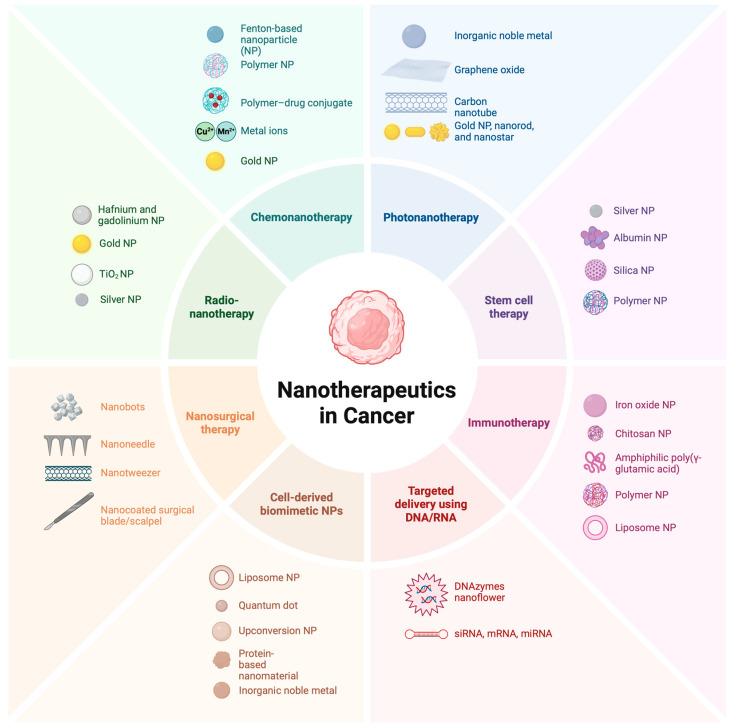
Nanomaterials as precision medicine in cancer treatment. A high-level overview of the potential uses of nanoparticles (NPs) in precision medicine (outside ring) and the biological obstacles that NPs can circumvent (inside ring). Intelligent NP designs that boost delivery can potentially speed the clinical translation of precision medicines by improving their performance, as addressed in this review. RNP is for ribonucleoprotein; CAR stands for chimeric antigen receptor; EGFR stands for epidermal growth factor receptor; EPR stands for improved permeability and retention.

**Figure 3 pharmaceutics-16-01076-f003:**
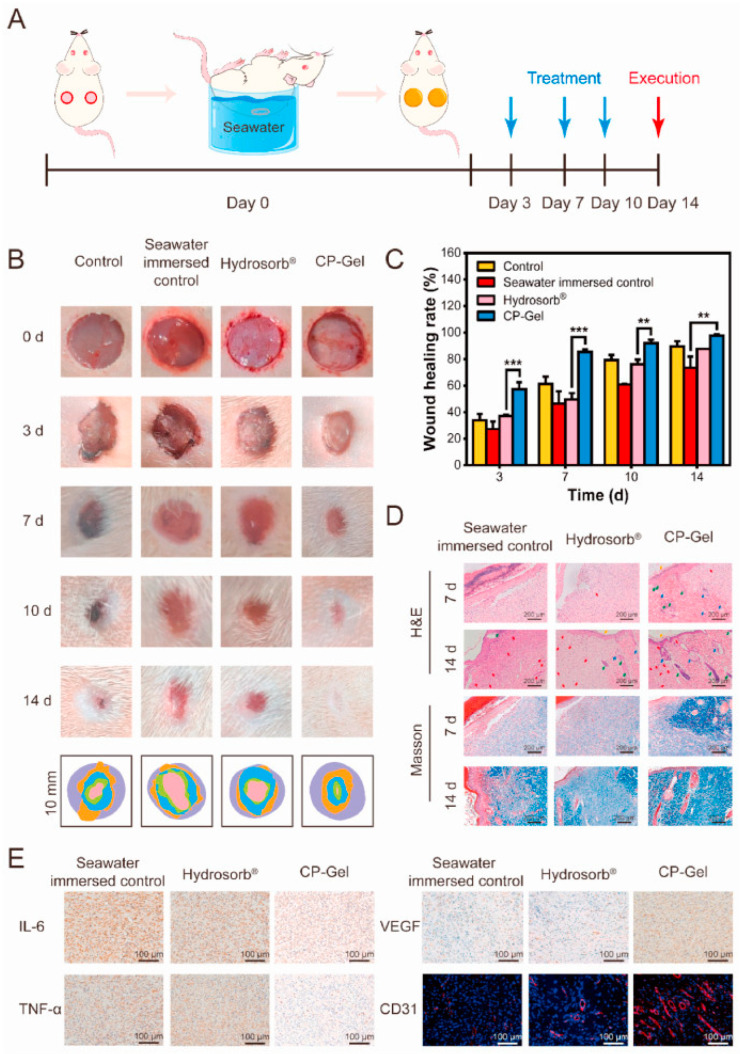
Wound healing in a model immersed in seawater using a chitosan-phenoxyethyl acrylate hydrogel. (**A**) A schematic showing the rat model of an experimental whole-layer wound immersed in seawater. (**B**) Images of wounds taken at 0, 3, 7, 10, and 14 days under various treatment conditions. (**C**) Rate of wound healing in various treatment groups at 0, 3, 7, 10, and 14 days. (**D**) Images of wounds at 7 and 14 days after treatment using H&E and Masson staining in various treatment groups. There are neovascular sebaceous glands (blue arrows), hair follicles (green arrows), and neovascular epithelia (yellow arrows) represented by the neovascularisation (red arrows). Masson staining photos reveal blue collagen. (**E**) Comparing various treatment groups at 7 days using immunohistochemistry for IL-6, TNF-α, VEFG, and immunofluorescence staining of CD31. Data are expressed as mean ± SD (*n* = 3). *** *p <* 0.001, ** *p <* 0.01. Adapted from Zhao et al., 2024 [97].

**Figure 4 pharmaceutics-16-01076-f004:**
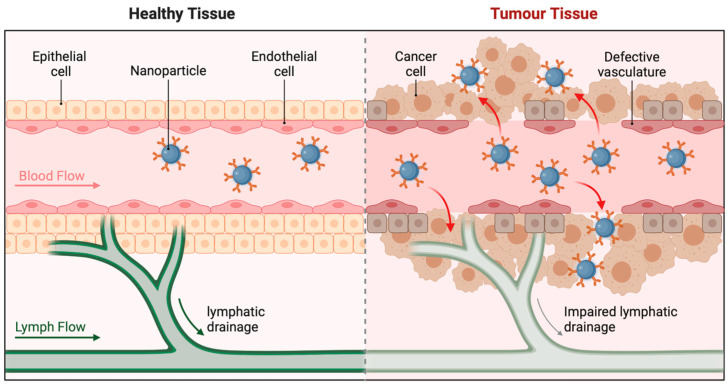
Passive targeting of nanoparticles to cancer cells. Nanoparticles penetrates the skin layer and target to tumour cells and thus, clear lymphatic drainage.

**Figure 5 pharmaceutics-16-01076-f005:**
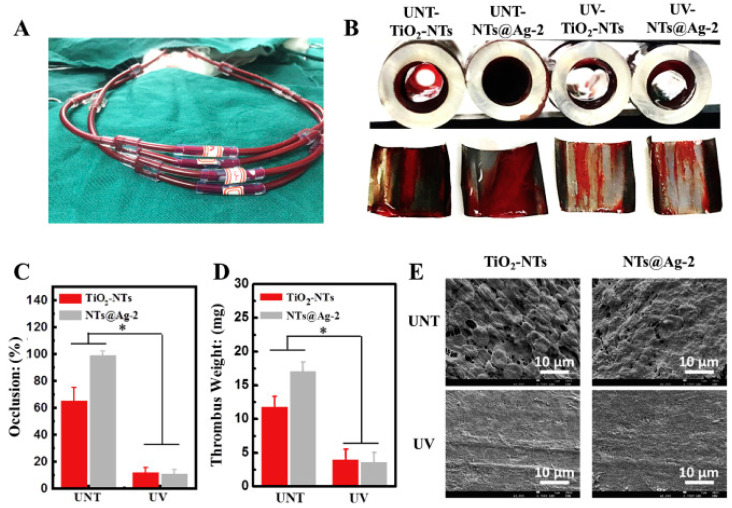
Improved vascular biocompatibility achieved by TiO_2_ nanotubes decorated with multifunctional Ag nanoparticles. Characterisation of ex vivo blood compatibility. The anticoagulant characteristics were evaluated ex vivo, as shown in photo (**A**). (**B**) Cross-sectional photos of the sample-containing catheters are shown after 30 min of circulation, with the thrombus on the sample surfaces visible at the front. This is the occlusion rate of the samples. (**C**) Occlusion rate of samples. (**D**) The samples of thrombosis were weighed. These are scanning electron micrographs of the samples (**E**). Adopted from Chen et al. (2021) [117]. * *p* < 0.05.

**Figure 6 pharmaceutics-16-01076-f006:**
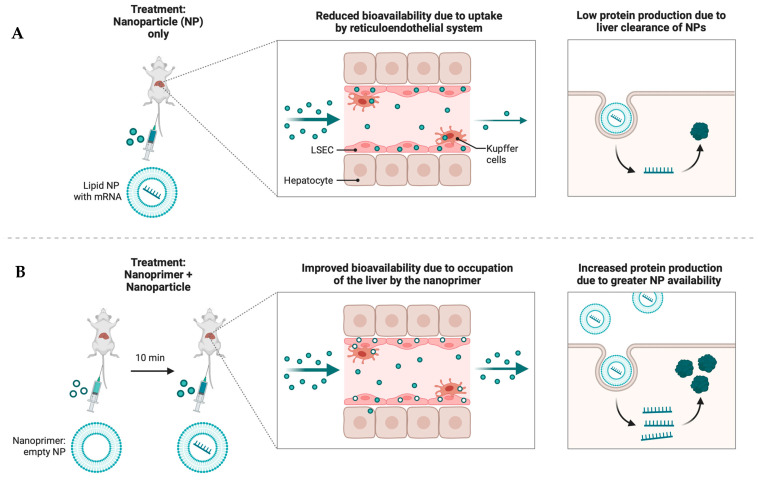
Nanoprimer to improve systemic delivery and biodistribution of nanoparticles. (**A**) Treatment with nanoparticle gives low protein production due to liver clearance of NPs. (**B**) Treatment with nanoprimer with nanoparticle gives increased protein production due to greater NP availability.

**Figure 7 pharmaceutics-16-01076-f007:**
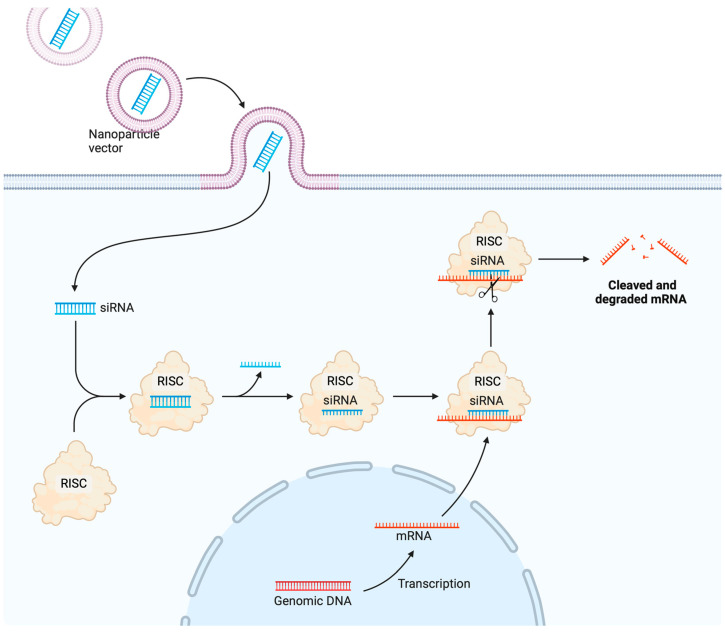
Small interfering RNA (siRNA) nanoparticle delivery system.

**Figure 8 pharmaceutics-16-01076-f008:**
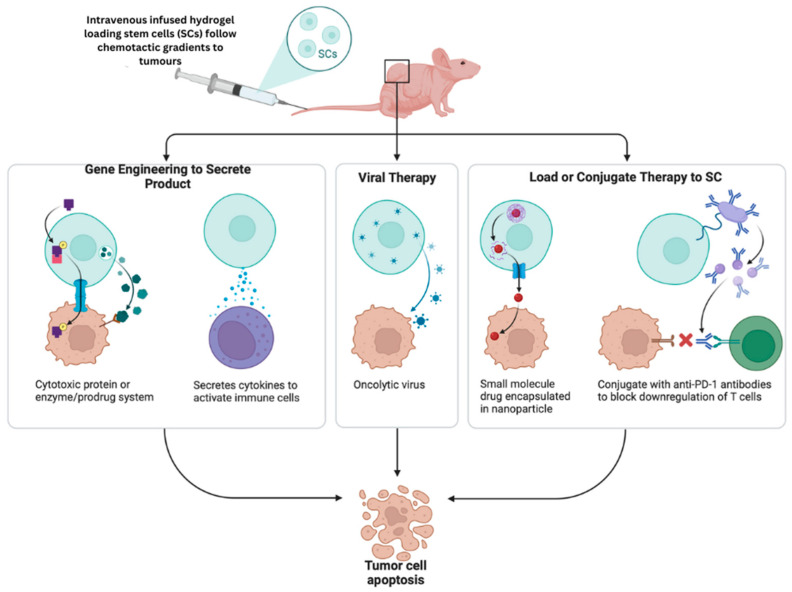
Schematic representation of migratory stem cell therapeutics against cancer cells. Using biomaterials in stem cell treatment, controlling the 3D cell culture’s changeable matrix characteristics controls differentiation, proliferation, migration, adhesion, and attachment. A spatiotemporally controlled 3D culture with customisable characteristics can simulate the native ECM, to guide stem cell behaviour to generate organoids or organs-on-chips. Organoids and OOCs are used for disease modelling, drug screening, etc. These 3D culture features can be used to build injectable hydrogels and stem cell patches for stem cell delivery. Low invasiveness, great efficiency, and reduced cell death are the advantages of these cell transport vehicles.

**Figure 9 pharmaceutics-16-01076-f009:**
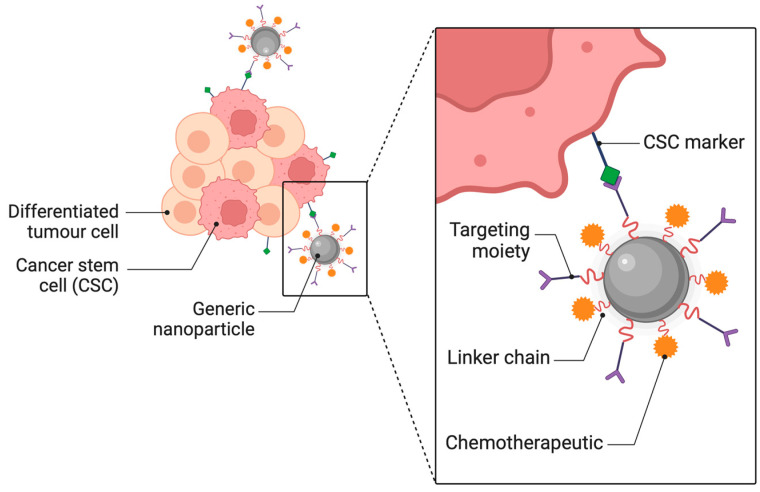
Active targeting of cancer stem cells with nanoparticles.

**Table 2 pharmaceutics-16-01076-t002:** Clinical studies on vaccines derived from VLPs.

Vaccine	Phase	Conditions	Project No.
VBI-2902a	I/II	COVID-19	NCT04773665
COVID-19	II/III	SARS-CoV-2 infection	NCT04662697
9-Valent HPV Vaccine	III	HPV/cervical/vulvar/vaginal/genital warts	NCT00543543
CHIKV VLP Vaccine	II	Chikungunya	NCT03483961
NoV GI.1/GII.4 Bivalent VLP vaccine	II	Norovirus	NCT02153112
V501	III	Cervical cancer/genital warts	NCT00092534
Quadrivalent VLP Influenza Vaccine	III	RNA virus/respiratory tract infections/virus illnesses	NCT03301051
Novartis Meningococcal ACWY Conjugate Vaccine	III	Meningitis/HPV/pertussis/tetanus	NCT00518180
RSV-F Protein Nanoparticle vaccine	II	RSV-F	NCT01960686
A/H1N1 2009 Influenza VLP vaccine	II	Seasonal influenza	NCT01072799
CYT006-AngQb	II	Mild essential hypertension/moderate essential hypertension	NCT00710372
Formalin-inactivated EV71 vaccine	III	HFMD	NCT01569581
VRC-ZKADNA090-00-VP	II	Infections include Zika, Virus, Flavivirus, Flaviviridae, and RNA viruses.	NCT03110770
HIV p17	I	HIV infections	NCT00001053

**Table 3 pharmaceutics-16-01076-t003:** Commercially accessible bio-material-based products.

Biomaterials	License Year	Commercial Name	Treatment	Indications
PLGA	1999	Neutropin subcutaneous	Microspheres	multiforme growth hormone deficiency
Lipid nanoparticles	1995	Doxil^®^	Chemotherapy (Doxorubicin)	multiple myeloma; breast neoplasms; ovarian neoplasms; Kaposi’s sarcoma
PCPP-SA	1996	Gliadel^®^	Intracranial wafer	
Poly (L-Lactide)	2016	Absorb GT1	Vascular scaffold system	coronary artery syndrome
Virus-like particle	2006	Depot^®^ Gardasil^®^	Vaccine delivery	cervical cancer
Polyethylene glycol-based	2018	Dextenza^®^	Insert (Dexamethasone)	amyloidosis in adults, ocular inflammation and pain
Lipid nanoparticles	2018	Onpattro^®^	Gene therapy (siRNA)	hereditary transthyretin-mediated polyneuropathy
PLGA microspheres	1989	Leupron Depot^®^	Hormone therapy	advanced prostate cancer
Hydrogel conjugated with fluorescein lipid nanoparticles	2020	Comirnaty^®^	Vaccine delivery	surgery SARS-CoV-2 infection
Crosslinked polydimethylsiloXane	1990	Norplant^®^ implant	Implant (Levonorgestrel)	contraceptive

**Table 4 pharmaceutics-16-01076-t004:** List of FDA-approved DDSs for drugs.

Drug	Date of First Approval	Application
Lipid-based		
DaunoXome	1996	Kaposi’s sarcoma
Doxil	1995	Ovarian cancer, Kaposi’s sarcoma, multiple myeloma
Visudyne	2000	myopia, wet age-related macular degeneration, ocular histoplasmosis
AmBisome	1997	Fungal/protozoal infections
Onivyde	2015	Metastatic pancreatic cancer
Marqibo	2012	Acute lymphoblastic leukaemia
Onpattro	2018	Transthyretin-mediated amyloidosis
Vyxeos	2017	Acute myeloid leukaemia
Polymer-based		
Copaxone	1996	Multiple sclerosis
Oncaspar	1994	Acute lymphoblastic leukaemia
Eligard	2002	Prostate cancer
PegIntron	2001	Hepatitis C infection
Abraxane	2005	Metastatic breast cancer, lung cancer, metastatic pancreatic cancer
Neulasta	2002	Neutropenia, chemotherapy induced
Plegridy	2014	Multiple sclerosis
Cimiza	2008	Rheumatoid arthritis, Crohn’s disease, psoriatic arthritis, ankylosing spondylitis
Adynovate	2015	Haemophilia
Inorganic		
DexFerrum	1996	Iron-deficient anaemia
INFeD	1992	Iron-deficient anaemia
Venofer	2000	Iron deficiency in chronic kidney disease
Ferrlecit	1999	Iron deficiency in chronic kidney disease
Injectafer	2013	Iron-deficient anaemia
Feraheme	2009	Iron deficiency in chronic kidney disease

## Data Availability

Data can be made available on request to corresponding authors.

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
