# Peer review of "Recent Review on Biological Barriers and Host–Material Interfaces in Precision Drug Delivery: Advancement in Biomaterial Engineering for Better Treatment Therapies"

_pharmaceutics, 2024, doi:10.3390/pharmaceutics16081076_

Round 1

Reviewer 1 Report

Comments and Suggestions for Authors

The authors seem to have summarized well the recent research trends on biological barriers and host-material interfaces in precision drug delivery.

1. In Figure 2 and Figure 9: Please correct any letters that are cut off or incorrectly placed.

2. L142: Change ‘the most. two separate sources.’ to ‘the most, two separate sources.’

3. In Figure 5: Change ‘TiO2’ to ‘TiO2’.

4. Throughout the manuscript: Please check again for typos and awkward phrases.

------------------------------------------------The end----------------------------------------

Author Response

Reviewer 1:

Comments 1:  In Figure 2 and Figure 9: Please correct any letters that are cut off or incorrectly placed.

Response 1: Thank you for pointing this out. We agree with this comment. Therefore, we have corrected figure 2 and figure 9 as suggested.

Comments 2:   L142: Change ‘the most. two separate sources.’ to ‘the most, two separate sources.’

Response 2: Thanks for your valuable suggestion. We have made the necessary changes .

Comments 3:   In Figure 5: Change ‘TiO2’ to ‘TiO2’.

Response 3: Thanks for your valuable suggestion. We have made the necessary changes in figure 5.

Comments 4 Throughout the manuscript: Please check again for typos and awkward phrases.

Response 4: Thanks for your kind observation and suggestion. The whole manuscript has been checked thoroughly for language editing and proof reading. The manuscript has been checked by professional native english writer and also certificate has given for the same’.

Response 4: Agreed. EPR challenges and future scope for drug development targeting has been discussed.

Comments 5:      For active targeting, it is more important to discuss transcytosable nanomedicine (transcytosis for crossing vasculature), not just cellular targeting. In fact, transcytosis into tumor has been proposed in the early developmental phases of drug delivery by Dvorak (Vascular permeability factor/vascular endothelial growth factor, microvascular hyperpermeability, and angiogenesis, Am J Pathol. 1995 May; 146(5): 1029–1039).

Response 5: Thanks for your kind observation and suggestion.  Active targeting of transcytosable nanomedicine has been added and cite the suggested work. Page 24, line 1066-1078. Reference [262]

Reviewer 2 Report

Comments and Suggestions for Authors

 This is an interesting review and the authors have collected a good dataset. However, the paper is generally poorly written (especially in an academic context). Reading through the manuscript can be an incredibly frustrating experience, particularly vagueness is an especially common problem (in terms of professional English writing). Further, the quality of artwork (graphs) are so bad. Here are some specific comments:

-       Page2, line62-64: Sentence “There are two main types of DDS systems: those that help with the spatial guidance of drug molecules to specific organs or diseased areas, and those that help with the better management of the temporal elements of drug activation”. Meaning is obscure, please re-write and be more specific.

-       Page2, line65-66: replace with “depending on the mode of administration, medicines or drug-delivery carriers may encounter various biological obstacles”.

-       Page2. Line67-75: The paragraph is hard to understand. I suggest major re-write of this paragraph.

-       Figure 1, Caption: The graph shows barriers encountered by nanoparticles but not ways to overcome these barriers, so what is meant by “overcoming”?

-       Figure 2: The figure presented is useless in its current form. Text-in-graph is corrupted. The figure should be clearly re-drawn.

-       Table 1: The column header “Target” does not seem correct. It looks like it is the drug molecule (targeted molecule) rather than target. Please specify if this statement is correct. 

-       Figure 3E: very bad resolution and hard to see. Please omit or replace with better resolution.

-       Table 4: Are these “List of FDA approved drugs” or “List of FDA approved DDS”?

-       Please check the references (100-104):

100. Yun YH, Lee BK, Park K, J. Controlled Release 2015, 219, 2. 

101. Allen TM, Cullis PR, Science 2004, 303, 1818. [PubMed: 15031496] 

102. Langer R, Nature 1998, 392, 5. [PubMed: 9579855] 

103. Langer R, Science 1990, 249, 1527. [PubMed: 2218494] 

104. Hoare TR, Kohane DS, Polymer 2008, 49, 1993. 

-       Reference 105 is confusing. Please check

a). Ensign LM, Cone R, Hanes J, Adv. Drug Delivery Rev 2012, 64, 557;b)Zhang S, Bellinger AM, Glettig DL, Barman R, Lee 1403 YA, Zhu J, Cleveland C, Montgomery VA, Gu L, Nash LD, Maitland DJ, Langer R, Traverso G, Nat. Mater 2015, 14, 1065; 1404 [PubMed: 26213897] c)Lowman AM, Morishita M, Kajita M, Nagai T, Peppas NA, J. Pharm. Sci 1999, 88, 933; [PubMed: 1405 10479357] d)Jeong B, Bae YH, Lee DS, Kim SW, Nature 1997, 388, 860; [PubMed: 9278046] e)Guvendiren ML, Lu HD, 1406 Burdick JA, Soft Matter 2011, 8, 260. 

-       There are unusual large number of references (334 references). Reducing this number is preferable.

Comments on the Quality of English Language

English language used is non-scientific where meanings in several parts are obscure. English writing requires scientific editing by professional.

Author Response

Reviewer 2

Comments 1:       Page2, line62-64: Sentence “There are two main types of DDS systems: those that help with the spatial guidance of drug molecules to specific organs or diseased areas, and those that help with the better management of the temporal elements of drug activation”. Meaning is obscure, please re-write and be more specific.

Response 1: Thanks for your kind observation and suggestion. It has been corrected as “Drug delivery systems (DDS) are designed to temporally and spatially control drug availability and activity. They assist in improving the balance between on-target therapeutic efficacy and off-target toxic side effects.”

Comments 2:       Page2, line65-66: replace with “depending on the mode of administration, medicines or drug-delivery carriers may encounter various biological obstacles”.

Response 2: Agree. We have corrected the sentence .

Comments 3:      Page2. Line67-75: The paragraph is hard to understand. I suggest major re-write of this paragraph.

Response 3: Thanks for your kind observation and suggestion. The paragraph has been revised as suggested.

Comments 4:       Figure 1, Caption: The graph shows barriers encountered by nanoparticles but not ways to overcome these barriers, so what is meant by “overcoming”?

Response 4: Thanks for your kind observation and suggestion. Figure 1 just defining various biological barriers present in the body for foreign particles to enter. However, legend for figure 1 has been revised as “Schematic representation of biological barriers that nanoparticles can help overcome.”

Comments 5:   Figure 2: The figure presented is useless in its current form. Text-in-graph is corrupted. The figure should be clearly re-drawn.

Response 5: Thanks for your kind observation and suggestion. We have redrawn the figure 2 to make text clear.

Comments 6:    Table 1: The column header “Target” does not seem correct. It looks like it is the drug molecule (targeted molecule) rather than target. Please specify if this statement is correct. 

Response 6: Agree. We have corrected to “targeted molecule”

Comments 7:     Figure 3E: very bad resolution and hard to see. Please omit or replace with better resolution.

Response 7 : Thanks for your kind observation and suggestion.  As the image was taken with permission and we have obtain high resolution image which has been incorporated in the manuscript.

Comments 8:     Table 4: Are these “List of FDA approved drugs” or “List of FDA approved DDS”?

Response 8: Thanks for your kind observation and suggestion. The legend of Table 4 has been corrected as “List of FDA approved DDS for the drugs”.

Comments 9:     Please check the references (100-104):

100. Yun YH, Lee BK, Park K, J. Controlled Release 2015, 219, 2. 

101. Allen TM, Cullis PR, Science 2004, 303, 1818. [PubMed: 15031496] 

102. Langer R, Nature 1998, 392, 5. [PubMed: 9579855] 

103. Langer R, Science 1990, 249, 1527. [PubMed: 2218494] 

104. Hoare TR, Kohane DS, Polymer 2008, 49, 1993. 

Response 9: Thanks for your kind observation and suggestion. We have solved the reference issue.

Comments 10:      Reference 105 is confusing. Please check

a). Ensign LM, Cone R, Hanes J, Adv. Drug Delivery Rev 2012, 64, 557;b)Zhang S, Bellinger AM, Glettig DL, Barman R, Lee 1403 YA, Zhu J, Cleveland C, Montgomery VA, Gu L, Nash LD, Maitland DJ, Langer R, Traverso G, Nat. Mater 2015, 14, 1065; 1404 [PubMed: 26213897] c)Lowman AM, Morishita M, Kajita M, Nagai T, Peppas NA, J. Pharm. Sci 1999, 88, 933; [PubMed: 1405 10479357] d)Jeong B, Bae YH, Lee DS, Kim SW, Nature 1997, 388, 860; [PubMed: 9278046] e)Guvendiren ML, Lu HD, 1406 Burdick JA, Soft Matter 2011, 8, 260. 

Response 10: Thanks for your kind observation and suggestion. We have solved the reference issue

Comments 11:      There are unusual large number of references. Reducing this number is preferable.

Response 11: Thanks for your kind observation and suggestion. We agree with your comments; however, we have cited many studies in this paper. We were tried to reduce the reference but still number is high.

Reviewer 3 Report

Comments and Suggestions for Authors The authors discussed biological barriers and host-material interaces in precision drug delivery. The authors try to cover everything in the manuscript. This leads to a bit of confusion in the structure of the article. I can see that the authors have put a lot of effort into this paper, particularly in discussing biology, and there is no doubt it has many useful things. With the respect to their efforts, I would like to offer some comments for the authors to consider. The authors should address the following concerns before publication. 1. It seems that almost all the figures are copied from other references. Please request reuse permissions. 2. In the introduction, the authors should discuss the logical relationship/connection between each section. Then the readers can easily follow the discussion. Two many things. 3. For escaping the reticuloendothelial system (RES) clearance, making stealth nanoparticle for long circulation becomes critical for subsequent targeting. Please check some useful references (such as https://doi.org/10.1016/j.addr.2023.114895). This is first major barriers for iv injection. 4. The issue of EPR has been discussed at some length by many in the literature, and I think the questions are quite complex. While I would not like to comment here on that topic, the very detailed literature, and controversy, is not mentioned. It might be suitable at least to alert readers to another layer of thinking. 5. For active targeting, it is more important to discuss transcytosable nanomedicine (transcytosis for crossing vasculature), not just cellular targeting. In fact, transcytosis into tumor has been proposed in the early developmental phases of drug delivery by Dvorak (Vascular permeability factor/vascular endothelial growth factor, microvascular hyperpermeability, and angiogenesis, Am J Pathol. 1995 May; 146(5): 1029–1039).

Author Response

Reviewer 3

Comments 1:      1. It seems that almost all the figures are copied from other references. Please request reuse permissions.

Response 11: Thanks for your kind observation and suggestion. Permission for figure 3 and figure 5 have been taken because we have used third party data. However, other figures were drawn originally after studying the paper. Still, we modify the current form of figures.

Comments 2:      In the introduction, the authors should discuss the logical relationship/connection between each section. Then the readers can easily follow the discussion. Two many things.

Response 2: Thanks for your kind observation and suggestion.  We have revised the introduction to connect the paragraphs.

Comments 3:      For escaping the reticuloendothelial system (RES) clearance, making stealth nanoparticle for long circulation becomes critical for subsequent targeting. Please check some useful references (such as https://doi.org/10.1016/j.addr.2023.114895). This is first major barriers for iv injection.

Response 3: Thanks for your kind observation and suggestion.  The suggested reference has been cited in the manuscript as ref. [196].

Comments 4:  The issue of EPR has been discussed at some length by many in the literature, and I think the questions are quite complex. While I would not like to comment here on that topic, the very detailed literature, and controversy, is not mentioned. It might be suitable at least to alert readers to another layer of thinking.

Response 4: Agreed. EPR challenges and future scope for drug development targeting has been discussed.

Comments 5:      For active targeting, it is more important to discuss transcytosable nanomedicine (transcytosis for crossing vasculature), not just cellular targeting. In fact, transcytosis into tumor has been proposed in the early developmental phases of drug delivery by Dvorak (Vascular permeability factor/vascular endothelial growth factor, microvascular hyperpermeability, and angiogenesis, Am J Pathol. 1995 May; 146(5): 1029–1039).

Response 5: Thanks for your kind observation and suggestion.  Active targeting of transcytosable nanomedicine has been added and cite the suggested work. Page 24, line 1066-1078. Reference [262]

Round 2

Reviewer 2 Report

Comments and Suggestions for Authors

1. Is the graph in Page 7 (in the revised manuscript) belong to Figure 2? or omitted and replaced with the one in Page 7? please remove it if intended to omit.

2. The same with the graph in Page 15. Is it belong or not to Figure 4? please remove it if intended to omit.

3. Figure 9 has two components. Please give them A and B, then explain them individually in the legend.

4. Table 3 title could be changed to "Commercially accessible bio-materials-based products".

Author Response

Comments 1:  ⁠Is the graph in Page 7 (in the revised manuscript) belong to Figure 2? or omitted and replaced with the one in Page 7? please remove it if intended to omit.

Response 1: Thanks for your valuable suggestion  Revised Figures was attached as Figure 2. Earlier one has deleted.

Comments 2:  .⁠ ⁠The same with the graph in Page 15. Is it belong or not to Figure 4? please remove it if intended to omit.

Response 2: Revised Figures was attached as Figure 4. Earlier one has deleted.

Comments 3: ⁠ ⁠Figure 9 (Its fig 6) has two components. Please give them A and B, then explain them individually in the legend.

Response 3: Thanks for your valuable suggestion. Agreed , we have labelled them as A and B, as per suggestion they have been explained individually in the legend

Comments 4:  ⁠Table 3 title could be changed to "Commercially accessible bio-materials-based products"

Response 4: Thanks for your valuable suggestion Table 3 title has been revised as per suggestion.

Reviewer 3 Report

Comments and Suggestions for Authors

The authors address the concerns well. I have no further comments.

Author Response

comments:The authors address the concerns well. I have no further comments.

Response: Thanks for your valuable suggestions